# SIC - MMAB: Synchronisation Involves Communication in Multiplayer Multi-Armed Bandits

**Etienne Boursier**
CMLA, ENS Paris-Saclay
`etienne.boursier@ens-paris-saclay.fr`

**Vianney Perchet**
CMLA, ENS Paris-Saclay
Criteo AI Lab, Paris
`vianney.perchet@normalesup.org`

## Abstract

Motivated by cognitive radio networks, we consider the stochastic multiplayer multi-armed bandit problem, where several players pull arms simultaneously and collisions occur if one of them is pulled by several players at the same stage. We present a decentralized algorithm that achieves the same performance as a centralized one, contradicting the existing lower bounds for that problem. This is possible by "hacking" the standard model by constructing a communication protocol between players that deliberately enforces collisions, allowing them to share their information at a negligible cost. This motivates the introduction of a more appropriate dynamic setting without sensing, where similar communication protocols are no longer possible. However, we show that the logarithmic growth of the regret is still achievable for this model with a new algorithm.

## 1 Introduction

In the stochastic Multi Armed Bandit problem (MAB), a single player sequentially takes a decision (or "pulls an arm") amongst a finite set of possibilities $[K] \coloneqq \{1, \ldots, K\}$. After pulling arm $k \in [K]$ at stage $t \in \mathbb{N}^*$, the player receives a random reward $X_k(t) \in [0, 1]$, drawn *i.i.d.* according to some unknown distribution $\nu_k$ of expectation $\mu_k \coloneqq \mathbb{E}[X_k(t)]$. Her objective is to maximize her cumulative reward up to stage $T \in \mathbb{N}^*$. This sequential decision problem, first introduced for clinical trials [27, 25], involves an "*exploration/exploitation* dilemma" where the player must trade-off acquiring *vs.* using information. The performance of an algorithm is controlled in term of **regret**, the difference of the cumulated reward of an optimal algorithm knowing the distributions $(\nu_k)_{k \in [K]}$ beforehand and the cumulated reward of the player. It is known that any "reasonable" algorithm must incur at least a logarithmic regret [19], which is attained by some existing algorithms such as UCB [1, 4].

MAB has been recently popularized thanks to its applications to online recommendation systems. Many different variants of MAB and classes of algorithms have thus emerged in the recent years [see 11]. In particular, they have been considered for cognitive radios [16], where the problem gets more intricate as multiple users are involved and they collide if they pull the same arm $k$ at the same time $t$, i.e., they transmit on the same channel. If this happens, they all receive $0$ as a reward instead of $X_k(t)$, meaning that no message is transmitted.

If a central agent controls simultaneously all players' behavior then a tight lower bound is known [3, 18]. Yet this centralized problem is not adapted to cognitive radios, as it allows communication between players at each time step; in practice, this induces significant costs in both energy and time. As a consequence, most of the current interest lies in the decentralized case [20, 2, 5], which presents another complication due to the feedback. Besides the received reward, an additional piece of information may be observed at each time step. When this extra observation is the collision indicator, Rosenski et al. [26] provided two algorithms for both a fixed and a varying number of players. They are based on a *Musical Chairs* procedure that quickly assigns players to different arms. Besson and

Kaufmann [8] provided an efficient UCB-based algorithm if $X_k(t)$ is observed instead[1]. Lugosi and Mehrabian [21] recently proposed an algorithm using no additional information. The performances of these algorithms and the underlying model differences are summarized in Table 1, Section 1.1.

The first non trivial lower bound for this problem has been recently improved [20, 8]. These lower bounds suggest that decentralization adds to the regret a multiplicative factor $M$, the number of players, compared to the centralized case [3]. Interestingly, these lower bounds scale linearly with the inverse of the gaps between the $\mu_k$ whereas this scaling is quadratic for most of the existing algorithms. This is due to the fact that although collisions account for most of the regret, lower bounds are proved without considering them.

Although it is out of our scope, the heterogeneous model introduced by Kalathil et al. [17] is worth mentioning. In this case, the reward distribution depends on each user [6, 7]. An algorithm reaching the optimal allocation without explicit communication between the players was recently proposed [9].

Our main contributions are the following:

**Section 2:** When collisions are observed, we introduce a new decentralized algorithm that is "hacking" the setting and induces communication between players through deliberate collisions. The regret of this algorithm reaches asymptotically (up to some universal constant) the lower bound of the centralized problem, meaning that the aforementioned lower bounds are unfortunately incorrect.

This algorithm relies on the unrealistic assumption that all users start transmitting at the very same time. It also explains why the current literature fails to provide near optimal results for the multiplayer bandits. It therefore appears that the assumption of synchronization has to be removed for practical application of the multiplayer bandits problem. On the other hand, this technique also shows that exhibiting lower bounds in multi-player MAB is more complex than in stochastic standard MAB.

**Section 3:** Without synchronization or collision observations, we propose the first algorithm with a logarithmic regret. The dependencies in the gaps between rewards yet become quadratic.

## 1.1 Models

In this section, we introduce different models of multiplayer MAB with a known number of arms $K$ but an unknown number of players $M \leq K$. The horizon $T$ is assumed known to the players (for simplicity of exposure, as the anytime generalization of results is now well understood [14]). At each time step $t \in [T]$, given their (private) information, all players $j \in [M]$ simultaneously pull the arms $\pi^j(t)$ and receive the reward $r^j(t) \in [0, 1]$ such that

$r^j(t) := X_{\pi^j(t)}(t)(1 - \eta_{\pi^j(t)}(t))$, where $\eta_{\pi^j(t)}(t)$ is the collision indicator defined by

$$\eta_k(t) := \mathbb{1}_{\#C_k(t) > 1} \qquad \text{with} \qquad C_k(t) := \{j \in [M] \mid \pi^j(t) = k\}.$$

The problem is **centralized** if players can communicate any information to each other. In that case, they can easily avoid collisions and share their statistics. In opposition, the problem is **decentralized** when players have only access to their own rewards and actions. The crucial concept we introduce is (a)synchronization between players. With synchronization, the model is called **static**.

**Assumption 1** (Synchronization). *Player $i$ enters the bandit game at the time $\tau_i = 0$ and stays until the final horizon $T$. This is common knowledge to all players.*

**Assumption 2** (Quasi-Asynchronization). *Players enter at different times $\tau_i \in \{0, \ldots, T-1\}$ and stay until the final horizon $T$. The $\tau_i$ are unknown to all players (including $i$).*

With quasi-asynchronicity[2], the model is **dynamic** and several variants already exist [26]. Denote by $\mathbf{M}(t)$ the set of players in the game at time $t$ (unknown but not random) and by $\mu_{(n)}$ the $n$-th order statistics of $\mu$, i.e., $\mu_{(1)} \geq \mu_{(2)} \geq \ldots \geq \mu_{(K)}$. The total regret is then defined for both static and dynamic models by:

$$R_T := \sum_{t=1}^{T} \sum_{k=1}^{\#\mathbf{M}(t)} \mu_{(k)} - \mathbb{E}_\mu \left[ \sum_{t=1}^{T} \sum_{j \in \mathbf{M}(t)} r^j(t) \right].$$

As mentioned in the introduction, different observation settings are considered.

**Collision Sensing:** Player $j$ observes $\eta_{\pi^j(t)}(t)$ and $r^j(t)$ at each time step.

**No sensing:** Player $j$ only observes $r^j(t)$, i.e., a reward of $0$ can indistinguishably come from a collision with another player or a null statistic $X_{\pi^j(t)}(t)$.

Notice that as soon as $\mathbb{P}(X_k = 0) = 0$, the No Sensing and Collision Sensing settings are equivalent. The setting where both $X_{\pi^j}(t)$ and $r^j(t)$ are observed is also considered in the literature and is called **Statistic Sensing** [8]. The No Sensing setting is the most difficult one as there is no extra observation.

Table 1 below compares the performances of the major algorithms, specifying the precise setting considered for each of them. The second algorithm of Lugosi and Mehrabian [21] and our algorithms also have problem independent bounds that are not mentioned in Table 1 for the sake of clarity. Due to space constraints, ADAPTED SIC-MMAB, SIC-MMAB2 and their related results are presented in Appendix C. Note that the two dynamic algorithms in Table 1 rely on different specific assumptions.

| Model | Algorithm's Reference | Prior knowledge | Asymptotic Upper bound (up to constant factor) |
|---|---|---|---|
| Centralized Multiplayer | Theorem 1 [18] | $M$ | $\sum_{k>M} \frac{\log(T)}{\mu_{(M)} - \mu_{(k)}}$ |
| Decentralized, Stat. Sensing | Theorem 11 [8] | $M$ | $M^3 \sum_{1 \leq i < k \leq K} \frac{\log(T)}{\left(\mu_{(i)} - \mu_{(k)}\right)^2}$ |
| Decentralized, Col. Sensing | Theorem 1 [26] | $\mu_{(M)} - \mu_{(M+1)}$ | $\frac{MK\log(T)}{\left(\mu_{(M)} - \mu_{(M+1)}\right)^2}$ |
| Decentralized, Col. Sensing | SIC-MMAB (Thm 1) | - | $\sum_{k>M} \frac{\log(T)}{\mu_{(M)} - \mu_{(k)}} + MK\log(T)$ |
| Decentralized, No Sensing | Theorem 1.1 [21] | $M$ | $\frac{MK\log(T)}{\left(\mu_{(M)} - \mu_{(M+1)}\right)^2}$ |
| Decentralized, No Sensing | Theorem 1.2 [21] | $M, \mu_{(M)}$ | $\frac{MK^2}{\mu_{(M)}} \log^2(T) + MK \frac{\log(T)}{\Delta'}$ |
| Decentralized, No Sensing | ADAPT. SIC-MMAB (Eq (13)) | $\mu_{(K)}$ | $\sum_{k>M} \frac{\log(T)}{\mu_{(M)} - \mu_{(k)}} + \frac{M^3 K \log(T)}{\mu_{(K)}} \log^2\left(\log(T)\right)$ |
| Decentralized, No Sensing | SIC-MMAB2 (Thm 3) | $\mu_{(K)}$ | $M \sum_{k>M} \frac{\log(T)}{\mu_{(M)} - \mu_{(k)}} + \frac{MK^2}{\mu_{(K)}} \log(T)$ |
| Dec., Col. Sensing , Dynamic | Theorem 2 [26] | $\bar{\Delta}_{(M)}$ | $\frac{M\sqrt{K\log(T)T}}{\bar{\Delta}_{(M)}^2}$ |
| Dec., No Sensing, Dynamic | DYN-MMAB (Thm 2) | - | $\frac{MK\log(T)}{\bar{\Delta}_{(M)}^2} + \frac{M^2 K \log(T)}{\mu_{(M)}}$ |

Table 1: Performances of different algorithms. Our algorithms and results are highlighted in red. $\bar{\Delta}_{(M)} \coloneqq \min_{i=1,\ldots,M}(\mu_{(i)} - \mu_{(i+1)})$ is the smallest gap among the top-$M+1$ arms and $\Delta' \coloneqq \min\{\mu_{(M)} - \mu_i \mid \mu_{(M)} - \mu_i > 0\}$ is the positive sub-optimality gap.

## 2 Collision Sensing: achieving centralized performances by communicating through collisions

In this section, we consider the Collision Sensing static model and prove that the decentralized problem is almost as complex, in terms of regret growth, as the centralized one. When players are synchronized, we provide an algorithm with an exploration regret similar to the known centralized lower bound [3]. This algorithm strongly relies on the synchronization assumption, which we leverage to allow communication between players through observed collisions. The communication protocol is detailed and explained in Section 2.2.3. This result also implies that the two lower bounds provided in the literature [8, 20] are unfortunately not correct. Indeed, the factor $M$ that was supposed to be the cost of the decentralization in the regret should not appear.

Let us now describe our algorithm SIC-MMAB. It consists of several phases.

1. The initialization phase first estimates the number of players and assigns ranks among them.

2. Players then alternate between exploration phases and communication phases.

(a) During the $p$-th exploration phase, each arm is pulled $2^p$ times and its performance is estimated in a Successive Accepts and Rejects fashion [22, 12].

(b) During the communication phases, players communicate their statistics to each other using collisions. Afterwards, the updated common statistics are known to all players.

3. The last phase, the exploitation one, is triggered for a player as soon as an arm is detected as optimal and assigned to her. This player then pulls this arm until the final horizon $T$.

## 2.1 Some preliminary notations

Players that are not in the exploitation phase are called **active**. We denote, with a slight abuse of notation, by $[M_p]$ the set of active players during the $p$-th phase of exploration-communication and by $M_p \leq M$ its cardinality. Notice that $M_p$ is non increasing because players never leave the exploitation phase.

Any arm among the top-$M$ ones is called **optimal** and any other arm is **sub-optimal**. Arms that still need to be explored (players cannot determine whether they are optimal or sub-optimal yet) are **active**. We denote, with the same abuse of notation, the set of active arms by $[K_p]$ of cardinality $K_p \leq K$. By construction of our algorithm, this set is common to all active players at each stage.

Our algorithm is based on a protocol called *sequential hopping* [15]. It consists of incrementing the index of the arm pulled by a specific player: if she plays arm $\pi_t^k$ at time $t$, she will play $\pi_{t+1}^k = \pi_t^k + 1 \pmod{[K_p]}$ at time $t+1$ during the $p$-th exploration phase.

## 2.2 Description of our protocol

As mentioned above, the SIC-MMAB algorithm consists of several phases. During the communication phase, players communicate with each other. At the end of this phase, each player thus knows the statistics of all players on all arms, so that this decentralized problem becomes similar to the centralized one. After alternating enough times between exploration and communication phases, sub-optimal arms are eliminated and players are fixed to different optimal arms and will exploit them until stage $T$. The complete pseudocode of SIC-MMAB is given in Algorithm 1, Appendix A.1.

### 2.2.1 Initialization phase

The objective of the first phase is to estimate the number of players $M$ and to assign **internal ranks** to players. First, players follow the Musical Chairs algorithm [26], described in Pseudocode 4, Appendix A.1, during $T_0 := \lceil K \log(T) \rceil$ steps in order to reach an **orthogonal setting**, i.e., a position where they are all pulling different arms. The index of the arm pulled by a player at stage $T_0$ will then be her **external rank**.

The second procedure, given by Pseudocode 5 in Appendix A.1, determines $M$ and assigns a unique internal rank in $[M]$ to each player. For example, if there are three players on arms 5, 7 and 2 at $t = T_0$, their external ranks are 5, 7 and 2 respectively, while their internal ranks are 2, 3 and 1. Roughly speaking, the players follow each other sequentially hopping through all the arms so that players with external ranks $k$ and $k'$ collide exactly after a time $k + k'$. Each player then deduces $M$ and her internal rank from observed collisions during this procedure that lasts $2K$ steps.

In the next phases, active players will always know the set of active players $[M_p]$. This is how the initial symmetry among players is broken and it allows the decentralized algorithm to establish communication protocols.

### 2.2.2 Exploration phase

During the $p$-th exploration phase, active players sequentially hop among the active arms for $K_p 2^p$ steps. Any active arm is thus pulled $2^p$ times by each active player. Using their internal rank, players start and remain in an orthogonal setting during the exploration phase, which is collision-free.

We denote by $B_s = 3\sqrt{\frac{\log(T)}{2s}}$ the error bound after $s$ pulls and by $T_k(p)$ (resp. $S_k(p)$) the centralized number of pulls (resp. sum of rewards) for the arm $k$ during the $p$ first exploration phases, i.e., $T_k(p) = \sum_{j=1}^{M} T_k^j(p)$ where $T_k^j(p)$ is the number of pulls for the arm $k$ by player $j$ during the $p$ first

exploration phases. During the communication phase, quantized rewards $\widetilde{S}_k^j(p)$ will be communicated between active players as described in Section 2.2.3.

After a succession of two phases (exploration and communication), an arm $k$ is **accepted** if

$$\#\left\{i \in [K_p] \,\big|\, \widetilde{\mu}_k(p) - B_{T_k(p)} \geq \widetilde{\mu}_i(p) + B_{T_i(p)}\right\} \geq K_p - M_p,$$

where $\widetilde{\mu}_k(p) = \frac{\sum_{m=1}^M \widetilde{S}_k^j(p)}{T_k(p)}$ is the centralized quantized empirical mean of the arm $k^3$, which is an approximation of $\hat{\mu}_k(p) = \frac{S_k(p)}{T_k(p)}$. This inequality implies that $k$ is among the top-$M_p$ active arms with high probability. In the same way, $k$ is **rejected** if

$$\#\left\{i \in [K_p] \,\big|\, \widetilde{\mu}_i(p) - B_{T_i(p)} \geq \widetilde{\mu}_k(p) + B_{T_k(p)}\right\} \geq M_p,$$

meaning that there are at least $M_p$ active arms better than $k$ with high probability. Notice that each player $j$ uses her own quantized statistics $\widetilde{S}_k^j(p)$ to accept/reject an arm instead of the exact ones $S_k^j(p)$. Otherwise, the estimations $\widetilde{\mu}_k(p)$ would indeed differ between the players as well as the sets of accepted and rejected arms. With Bernoulli distributions, the quantization becomes unnecessary and the confidence bound can be chosen as $B_s = \sqrt{2\log(T)/s}$.

### 2.2.3 Communication phase

In this phase, each active player communicates, one at a time, her statistics of the active arms to all other active players. Each player has her own communicating arm, corresponding to her internal rank. When the player $j$ is communicating, she sends a bit at a time step to the player $l$ by deciding which arm to pull: a 1 bit is sent by pulling the communicating arm of player $l$ (a collision occurs) and a 0 bit by pulling her own arm. The main originality of SIC-MMAB comes from this trick which allows implicit communication through collisions and is used in subsequent papers [13, 10, 24]. In an independent work, Tibrewal et al. [28] also proposed an algorithm using similar communication protocols for the heterogeneous case.

As an arm is pulled $2^n$ times by a single player during the $n$-th exploration phase, it has been pulled $2^{p+1} - 1$ times in total at the end of the $p$-th phase and the statistic $S_k^j(p)$ is a real number in $[0, 2^{p+1} - 1]$. Players then send a quantized **integer** statistic $\widetilde{S}_k^j(p) \in [2^{p+1} - 1]$ to each other in $p+1$ bits, i.e., collisions. Let $n = \lfloor S_k^j(p) \rfloor$ and $d = S_k^j(p) - n$ be the integer and decimal parts of $S_k^j(p)$, the quantized statistic is then $n + 1$ with probability $d$ and $n$ otherwise, so that $\mathbb{E}[\widetilde{S}_k^j(p)] = S_k^j(p)$.

An active player can have three possible statuses during the communication phase:

1. either she is receiving some other players' statistics about the arm $k$. In that case, she proceeds to **Receive Protocol** (see Pseudocode 1).

2. Or she is sending her quantized statistics about arm $k$ to player $l$ (who is then receiving). In that case, she proceeds to **Send Protocol** (see Pseudocode 2) to send them in a time $p + 1$.

3. Or she is pulling her communicating arm, while waiting for other players to finish communicating statistics among them.

Communicated statistics are all of length $p + 1$, even if they could be sent with shorter messages, in order to maintain synchronization among players. Using their internal ranks, the players can communicate in turn without interfering with each other. The general protocol for each communication phase is described in Pseudocode 3 below.

At the end of the communication phase, all active players know the statistics $\widetilde{S}_k^j(p)$ and so which arms to accept or reject. Rejected arms are removed right away from the set of active arms. Thanks to the assigned ranks, accepted arms are assigned to one player each. The remaining active players then update both sets of active players and arms as described in Algorithm 1, line 21.

This communication protocol uses the fact that a bit can be sent with a single collision. Without sensing, this can not be done in a single time step, but communication is still somehow possible. A bit can then be sent in $\frac{\log(T)}{\mu_{(K)}}$ steps with probability $1 - \frac{1}{T}$. Using this trick, two different algorithms relying on communication protocols are proposed in Appendix C for the No Sensing setting.

| **Receive Protocol** | **Send Protocol** |
|---|---|
| **Input:** $p$ (phase number), $l$ (own internal rank), $[K_p]$ (set of active arms) | **Input:** $l$ (player receiving), $s$ (statistics to send), $p$ (phase number), $j$ (own internal rank), $[K_p]$ (set of active arms) |
| **Output:** $s$ (statistic sent by the sending player) | |

**Receive Protocol**

**Input:** $p$ (phase number), $l$ (own internal rank), $[K_p]$ (set of active arms)
**Output:** $s$ (statistic sent by the sending player)

1: $s \leftarrow 0$ and $\pi \leftarrow$ index of the $l$-th active arm
2: **for** $n = 0, \dots, p$ **do**
3:    Pull $\pi$
4:    **if** $\eta_\pi(t) = 1$ **then**   # *other player sends* 1
5:      $s \leftarrow s + 2^n$ **end if**
6: **end for**
7: **return** $s$        # *sent statistics*

Pseudocode 1: receive statistics of length $p + 1$.

**Send Protocol**

**Input:** $l$ (player receiving), $s$ (statistics to send), $p$ (phase number), $j$ (own internal rank), $[K_p]$ (set of active arms)

1: $\mathbf{m} \leftarrow$ binary writing of $s$ of length $p + 1$, i.e., $s = \sum_{n=0}^{p} m_n 2^n$
2: **for** $n = 0, \dots, p$ **do**
3:    **if** $m_n = 1$ **then**
4:      Pull the $l$-th active arm    # *send* 1
5:    **else** Pull the $j$-th active arm    # *send* 0
6:    **end if**
7: **end for**

Pseudocode 2: send statistics $s$ of length $p + 1$ to player $l$.

---

**Communication Protocol**

**Input:** $\mathbf{s}$ (personal statistics of previous phases), $p$ (phase number), $j$ (own internal rank), $[K_p]$ (set of active arms), $[M_p]$ (set of active players)
**Output:** $\widetilde{\mathbf{S}}$ (quantized statistics of all active players)

1: For all $k$, sample $\widetilde{s}[k] = \begin{cases} \lfloor s[k] \rfloor + 1 \text{ with probability } s[k] - \lfloor s[k] \rfloor \\ \lfloor s[k] \rfloor \text{ otherwise} \end{cases}$     # *quantization*

2: Define $E_p := \{(i, l, k) \in [M_p] \times [M_p] \times [K_p] \mid i \neq l\}$ and set $\widetilde{\mathbf{S}}^{\mathbf{j}} \leftarrow \widetilde{\mathbf{s}}$
3: **for** $(i, l, k) \in E_p$ **do**     # *Player $i$ sends stats of arm $k$ to player $l$*
4:    **if** $i = j$ **then** Send $(l, \widetilde{s}[k], p, j, [K_p])$     # *player communicating*
5:    **else if** $l = j$ **then** $\widetilde{S}^i[k] \leftarrow$ Receive$(p, j, [K_p])$     # *player receiving*
6:    **else for** $p + 1$ time steps **do** Pull the $j$-th active arm **end for**   # *wait while others communicate*
7:    **end if**
8: **end for**
9: **return** $\widetilde{\mathbf{S}}$

Pseudocode 3: player with rank $j$ proceeds to the $p$-th communication phase.

### 2.2.4 Regret bound of SIC-MMAB

Theorem 1 bounds the expected regret incurred by SIC-MMAB. Due to space constraints, its proof is delayed to Appendix A.2.

**Theorem 1.** *With the choice $T_0 = \lceil K \log(T) \rceil$, for any given set of parameters $K$, $M$ and $\boldsymbol{\mu}$:*

$$\mathbb{E}[R_T] \leq c_1 \sum_{k > M} \min \left\{ \frac{\log(T)}{\mu_{(M)} - \mu_{(k)}}, \sqrt{T \log(T)} \right\} + c_2 K M \log(T)$$

$$+ c_3 K M^3 \log^2 \left( \min \left\{ \frac{\log(T)}{(\mu_{(M)} - \mu_{(M+1)})^2}, T \right\} \right)$$

*where $c_1$, $c_2$ and $c_3$ are universal constants.*

The first, second and third terms respectively correspond to the regret incurred by the exploration, initialization and communication phases, which dominate the regret due to low probability events of bad initialization or incorrect estimations. Notice that the minmax regret scales with $\mathcal{O}(K\sqrt{T \log(T)})$. Experiments on synthetic data are described in Appendix A.3. They empirically confirm that SIC-MMAB scales better than MCTopM [8] with the gaps $\Delta$, besides having a smaller minmax regret.

## 2.3 In contradiction with existing lower bounds?

Theorem 1 is in contradiction with the two existing lower bounds [8, 20], however SIC-MMAB respects the conditions required for both. It was thought that the decentralized lower bound was $\Omega \left( M \sum_{k > M} \frac{\log(T)}{\mu_{(M)} - \mu_{(k)}} \right)$, while the centralized lower bound was already known to be $\Omega \left( \sum_{k > M} \frac{\log(T)}{\mu_{(M)} - \mu_{(k)}} \right)$ [3]. However, it appears that the asymptotic regret of the decentralized case is not that much different from the latter, at least if players are synchronized. Indeed, SIC-MMAB takes advantage of this synchronization to establish communication protocols as players are able to

communicate through collisions. Subsequent papers [10, 24] recently improved the communication protocols of SIC-MMAB to obtain both initialization and communication costs constant in $T$, confirming that the lower bound of the centralized case is also tight for the decentralized model considered so far.

Liu and Zhao [20] proved the lower bound "by considering the best case that they do not collide". This is only true if colliding does not provide valuable information and the policies just maximize the losses at each round, disregarding the information gathered for the future. Our algorithm is built upon the idea that the value of the information provided by collisions can exceed in the long run the immediate loss in rewards (which is standard in dynamic programming or reinforcement learning for instance). The mistake of Besson and Kaufmann [8] is found in the proof of Lemma 12 after the sentence "We now show that second term in (25) is zero". The conditional expectation cannot be put inside/outside of the expectation as written and the considered term, which corresponds to the difference of information given by collisions for two different distributions, is therefore not zero. These two lower bounds disregarded the amount of information that can be deduced from collisions, while SIC-MMAB obviously takes advantage of this information.

Our exploration regret reaches, up to a constant factor, the lower bound of the centralized problem [3]. Although it is sub-logarithmic in time, the communication cost scales with $KM^3$ and can thus be predominant in practice. Indeed for large networks, $M^3$ can easily be greater than $\log(T)$ and the communication cost would then prevail over the other terms. This highlights the importance of the parameter $M$ in multiplayer MAB and future work should focus on the dependency in both $M$ and $T$ instead of only considering asymptotic results in $T$.

Synchronization is not a reasonable assumption for practical purposes and it also leads to undesirable algorithms relying on communication protocols such as SIC-MMAB. We thus claim that this assumption should be removed in the multiplayer MAB and the *dynamic model* should be considered instead. However, this problem seems complex to model formally. Indeed, if players stay in the game only for a very short period, learning is not possible. The difficulty to formalize an interesting and nontrivial dynamic model may explain why most of the literature focused on the static model so far.

## 3   Without synchronization, the dynamic setting

From now on, we no longer assume that players can communicate using synchronization. In the previous section, it was crucial that all exploration/communication phases start and end at the same time. This assumption is clearly unrealistic and should be alleviated, as radios do not start and end transmitting simultaneously. We also consider the more difficult No Sensing setting in this section.

We assume in the following that players do not leave the game once they have started. Yet, we mention that our results can also be adapted to the cases when players can leave the game during specific intervals or share an internal synchronized clock [26]. If the time is divided in several intervals, DYN-MMAB can be run independently on each of these intervals as suggested by Rosenski et al. [26]. In some cases, players will be leaving in the middle of these intervals, leading to a large regret. But for any other interval, every player stays until its end, thus satisfying Assumption 2.

In this section, Assumption 2 holds. At each stage $t = t_j + \tau_j$, player $j$ does not know $t$ but only $t_j$ (duration since joining). We denote by $T^j = T - \tau_j$ the (known) time horizon of player $j$.

### 3.1   A logarithmic regret algorithm

As synchronization no longer holds, we propose the DYN-MMAB algorithm, relying on different tools than SIC-MMAB. The main ideas of DYN-MMAB are given in Section 3.2. Its thorough description as well as the proof of the regret bound are delayed to Appendix B due to space constraints.

The regret incurred by DYN-MMAB in the dynamic No Sensing model is given by Theorem 2 and its proof is delayed to Appendix B.2. We also mention that DYN-MMAB leads to a Pareto optimal configuration in the more general problem where users' reward distributions differ [17, 6, 7, 9].

**Theorem 2.** *In the dynamic setting, the regret incurred by* DYN-MMAB *is upper bounded as follows:*

$$\mathbb{E}[R_T] \leq \mathcal{O}\left( \frac{M^2 K \log(T)}{\mu_{(M)}} + \frac{MK \log(T)}{\bar{\Delta}_{(M)}^2} \right),$$

*where $M = \#\mathbf{M}(T)$ is the total number of players in the game and $\bar{\Delta}_{(M)} = \min_{i=1,...,M}(\mu_{(i)} - \mu_{(i+1)})$.*

## 3.2 A communication-less protocol

DYN-MMAB's ideas are easy to understand but the upper bound proof is quite technical. This section gives some intuitions about DYN-MMAB and its performance guarantees stated in Theorem 2.

A player will only follow two different sampling strategies: either she samples uniformly at random in $[K]$ during the exploration phase; or she exploits an arm and pulls it until the final horizon. In the first case, the exploration of the other players is not too disturbed by collisions as they only change the mean reward of all arms by a common multiplicative term. In the second case, the exploited arm will appear as sub-optimal to the other players, which is actually convenient for them as this arm is now exploited.

During the exploration phase, a player will update a set of arms called Occupied $\subset [K]$ and an ordered list of arms called Preferences $\in [K]^\star$. As soon as an arm is detected as occupied (by another player), it is then added to Occupied (which is the empty set at the beginning). If an arm is discovered to be the best one amongst those that are neither in Occupied nor in Preferences, it is then added to Preferences (at the last position). An arm is **active** for player $j$ if it was neither added to Occupied nor to Preferences by this player yet.

To handle the fact that players can enter the game at anytime, we introduce the quantity $\gamma^j(t)$, the expected multiplicative factor of the means defined by

$$\gamma^j(t) = \frac{1}{t} \sum_{t'=1+\tau_j}^{t+\tau_j} \mathbb{E}\Big[(1 - \frac{1}{K})^{m_{t'}-1}\Big],$$

where $m_t$ is the number of players in their exploration phase at time $t$. The value of $\gamma^j(t)$ is unknown to the player and random but it only affects the analysis of DYN-MMAB and not how it runs.

The objective of the algorithm is still to form estimates and confidence intervals of the performances of arms. However, it might happen that the true mean $\mu_k$ does not belong to this confidence interval. Indeed, this is only true for $\gamma^j(t)\mu_k$, if the arm $k$ is still free (not exploited). This is the first point of Lemma 1 below. Notice that as soon as the confidence interval for the arm $i$ dominates the confidence interval for the arm $k$, then it must hold that $\gamma^j(t)\mu_i \geq \gamma^j(t)\mu_k$ and thus arm $i$ is better than $k$.

The second crucial point is to detect when an arm $k$ is exploited by another player. This detection will happen if a player receives too many 0 rewards successively (so that it is statistically very unlikely that this arm is not occupied). The number of zero rewards needed for player $j$ to disregard arm $k$ is denoted by $L_k^j$, which is sequentially updated during the process (following the rule of Equation (4) in Appendix B.1), so that $L_k^j \geq 2e \log(T^j)/\mu_k$. As the probability of observing a 0 reward on a free arm $k$ is smaller than $1 - \mu_k/e$, no matter the current number of players, observing $L_k^j$ successive 0 rewards on an unexploited arm happens with probability smaller than $\frac{1}{(T^j)^2}$.

The second point of Lemma 1 then states that an exploited arm will either be quickly detected as occupied after observing $L_k^j$ zeros (if $L_k^j$ is small enough) or its average reward will quickly drop because it now gives zero rewards (and it will be dominated by another arm after a relatively small number of pulls). The proof of Lemma 1 is delayed to Appendix B.2.

**Lemma 1.** *We denote by $\hat{r}_k^j(t)$ the empirical average reward of arm $k$ for player $j$ at stage $t + \tau_j$.*

1. *For any player $j$ and arm $k$, if $k$ is still free at stage $t + \tau_j$, then*

$$\mathbb{P}\Big[|\hat{r}_k^j(t) - \gamma^j(t)\mu_k| > 2\sqrt{\frac{6\,K\log(T^j)}{t}}\Big] \leq \frac{4}{(T^j)^2}.$$

   *We then say that the arm $k$ is **correctly estimated** by player $j$ if $|\hat{r}_k^j(t) - \gamma^j(t)\mu_k| \leq 2\sqrt{\frac{6\,K\log(T^j)}{t}}$ holds as long as $k$ is free.*

2. *On the other hand, if $k$ is exploited by some player $j' \neq j$ at stage $t^0 + \tau_j$, then, conditionally on the correct estimation of all the arms by player $j$, with probability $1 - \mathcal{O}\left(\frac{1}{T^j}\right)$:*
   * *either $k$ is added to Occupied at a stage at most $t^0 + \tau_j + \mathcal{O}\left(\frac{K\log(T)}{\mu_k}\right)$ by player $j$,*
   * *or $k$ is dominated by another unoccupied arm $i$ (for player $j$) at stage at most $\mathcal{O}\left(\frac{K\log(T)}{\mu_i^2}\right) + \tau_j$.*

It remains to describe how players start exploiting arms. After some time (upper-bounded by Lemma 10 in Appendix B.2), an arm which is still free and such that all better arms are occupied will be detected as the best remaining one. The player will try to occupy it, and this happens as soon as she gets a positive reward from it: either she succeeds and starts exploiting it, or she fails and assumes it is occupied by another player (this only takes a few number of steps, see Lemma 1). In the latter case, she resumes exploring until she detects the next available best arm. With high probability, the player will necessarily end up exploiting an arm while all the better arms are already exploited by other players.

## 4 Conclusion

We have presented algorithms for different multiplayer bandits models. The first one illustrates why the assumption of synchronization between the players is basically equivalent to allowing communication. Since communication through collisions is possible with other players at a sub-logarithmic cost, the decentralized multiplayer bandits is almost equivalent to the centralized one for the considered model. However, this communication cost has a large dependency in the number of agents in the network. Future work should then focus on considering both the dependency in time and the number of players as well as developing efficient communication protocols.

Our major claim is that synchronization should not be considered anymore, unless communication is allowed. We thus introduced a dynamic model and proposed the first algorithm with a logarithmic regret.

### Acknowledgments

This work was supported in part by a public grant as part of the Investissement d'avenir project, reference ANR-11-LABX-0056-LMH, LabEx LMH, in a joint call with Gaspard Monge Program for optimization, operations research and their interactions with data sciences.

## Footnotes

[1] We stress that $X_k(t)$ does not necessarily correspond to the received reward in case of collision.

[2] We prefer not to mention asynchronicity as players still use shared discrete time slots.

[3]For a player $j$ already exploiting since the $p^j$-th phase, we instead use the last statistic $\widetilde{S}_k^j(p) = \widetilde{S}_k^j(p^j)$.

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
