[Supplementary Material · SIC-MMAB-full.pdf]

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

[4]The code is available at `https://github.com/eboursier/sic-mmab`.

[5]Actually, a lower bound of $\mathbb{P}[X_k > 0]$ is enough. We instead use $\mu_{\min}$, as $\mathbb{P}[X_k > 0] \geq \mu_k$.

[6]The length of the Musical chairs and the estimation protocol in the initialization will also be respectively multiplied by $\frac{1}{\mu_{\min}}$ and $\frac{\log(T)}{\mu_{\min}}$.

[7]Of course, she does not declare any arm previously declared by another player.

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

# A Complementary material for Section 2

## A.1 Algorithm description

We here describe in detail the SIC-MMAB algorithm. All the pseudocodes are described from the point of view of a single player, which is the natural way to describe a decentralized algorithm. First, this algorithm relies on the Musical Chairs algorithm, introduced by Rosenski et al. [26]. We recall it in Pseudocode 4.

---

**MusicalChairs Protocol**

    **Input:** $[K_p]$ (set of active arms), $T_0$ (time of procedure)
    **Output:** Fixed (external rank)

1:  Initialize Fixed $\leftarrow -1$
2:  **for** $T_0$ time steps **do**
3:     **if** Fixed $= -1$ **then**
4:        Sample $k$ uniformly at random in $[K_p]$ and play it in round $t$
5:        **if** $\eta_k(t) = 0$ ($r_k(t) > 0$ for No Sensing setting) **then**
6:           Fixed $\leftarrow k$ **end if**                                *# The player stays in arm $k$ if no collision*
7:     **else** Play Fixed **end if**
8:  **end for**
9:  **return** Fixed                                              *# External rank*

---

Pseudocode 4: reach an orthogonal setting in $T_0$ steps.

The initialization phase then consists of a second procedure. Its purpose is to estimate $M$ and to assign different ranks in $[M]$ to all players. This procedure is described in Pseudocode 5 below. SIC-MMAB is finally described in Algorithm 1.

---

**Estimate_M Protocol**

    **Input:** $k \in [K]$ (external rank)
    **Output:** $M$ (estimated number of players), $j$ (internal rank)

1:  Initialize $M \leftarrow 1$, $j \leftarrow 1$ and $\pi \leftarrow k$                 *# estimates of $M$ and the internal rank*
2:  **for** $2k$ time steps **do**
3:     Pull $\pi$;     **if** $\eta_\pi(t) = 1$ **then** $M \leftarrow M + 1$ and $j \leftarrow j + 1$ **end if**     *# increases if collision*
4:  **end for**
5:  **for** $2(K - k)$ time steps **do**
6:     $\pi \leftarrow \pi + 1 \pmod{K}$ and pull $\pi$                           *# sequential hopping*
7:     **if** $\eta_\pi(t) = 1$ **then** $M \leftarrow M + 1$ **end if**                      *# increases if collision*
8:  **end for**
9:  **return** $M, j$

---

Pseudocode 5: estimate $M$ and assign ranks to the players.

---

**Algorithm 1** SIC-MMAB algorithm

**Input:** $T$ (horizon)
1: **Initialization Phase:**
2: Initialize Fixed $\leftarrow -1$ and $T_0 \leftarrow \lceil K \log(T) \rceil$
3: $k \leftarrow$ MusicalChairs $([K], T_0)$
4: $(M, j) \leftarrow$ Estimate_M $(k)$          *# estimated number of players and assigned internal rank*
5: Initialize $p \leftarrow 1$; $M_p \leftarrow M$; $[K_p] \leftarrow [K]$ and $\widetilde{\mathbf{S}}, \mathbf{s}, \mathbf{T} \leftarrow$ Zeros$(K)$     *# Zeros(K) returns a*
                                                    *vector of length $K$ containing only zeros*

6: **while** Fixed $= -1$ **do**

7:     **Exploration Phase:**
8:     $\pi \leftarrow j$-th active arm                                              *# start of a new phase*
9:     **for** $K_p 2^p$ time steps **do**
10:        $\pi \leftarrow \pi + 1 \pmod{[K_p]}$ and play $\pi$ in round $t$              *# sequential hopping*
11:        $s[\pi] \leftarrow s[\pi] + r_\pi(t)$                          *# Update individual statistics*
12:     **end for**

13:     **Communication Phase:**
14:     $\widetilde{\mathbf{S}}_{\mathbf{p}} \leftarrow$ Communication$(\mathbf{s}, p, j, [K_p], [M_p])$ and $\widetilde{\mathbf{S}}^{\mathbf{l}} \leftarrow \widetilde{\mathbf{S}}_{\mathbf{p}}^{\mathbf{l}}$ for every active player $l$
15:     $T[k] \leftarrow T[k] + M_p 2^p$ for every active arm $k$

16:     **Update Statistics:**                                        *# recall that $B_s = 3\sqrt{\frac{\log(T)}{2s}}$ here*

17:     Rej $\leftarrow$ set of active arms $k$ verifying $\#\left\{i \in [K_p] \,\Big|\, \frac{\sum_{l=1}^{M} \widetilde{S}^l[i]}{T[i]} - B_{T[i]} \geq \frac{\sum_{l=1}^{M} \widetilde{S}^l[k]}{T[k]} + B_{T[k]}\right\} \geq M_p$

18:     Acc $\leftarrow$ set of active arms $k$ verifying $\#\left\{i \in [K_p] \,\Big|\, \frac{\sum_{l=1}^{M} \widetilde{S}^l[k]}{T[k]} - B_{T[k]} \geq \frac{\sum_{l=1}^{M} \widetilde{S}^l[i]}{T[i]} + B_{T[i]}\right\} \geq$
        $K_p - M_p$, ordered according to their indices
19:     **if** $M_p - j + 1 \leq$ length(Acc) **then** Fixed $\leftarrow$ Acc$[M_p - j + 1]$        *# Start exploiting*
20:     **else**                                                  *# Update all the statistics*
21:        $M_p \leftarrow M_p -$ length(Acc) and $[K_p] \leftarrow [K_p] \setminus ($Acc $\cup$ Rej$)$
22:     **end if**
23:     $p \leftarrow p + 1$
24: **end while**

25: **Exploitation Phase:** Pull Fixed until $T$

---

## A.2 Regret analysis of SIC-MMAB

In this section, we prove the regret bound for SIC-MMAB algorithm given by Theorem 1. In what follows, the statement *"with probability $1 - \mathcal{O}(\delta(T))$, it holds that $f(T) = \mathcal{O}(g(T))$"* means that there is a universal constant $c \in \mathbb{R}_+$ such that $f(T) \leq cg(T)$ with probability at least $1 - c\delta(T)$.

We first decompose the regret as follows:

$$R_T = R^{\text{init}} + R^{\text{comm}} + R^{\text{explo}}, \tag{1}$$

where
$$
\begin{cases}
R^{\text{init}} = T_{\text{init}} \sum_{k=1}^{M} \mu_{(k)} - \mathbb{E}_\mu\Big[\sum_{t=1}^{T_{\text{init}}} \sum_{j=1}^{M} r^j(t)\Big] \text{ with } T_{\text{init}} = T_0 + 2K, \\[2ex]
R^{\text{comm}} = \mathbb{E}_\mu\Big[\sum_{t \in \text{Comm}} \sum_{j=1}^{M} (\mu_{(j)} - r^j(t))\Big] \text{ with Comm the set of communication steps,} \\[2ex]
R^{\text{explo}} = \mathbb{E}_\mu\Big[\sum_{t \in \text{Explo}} \sum_{j=1}^{M} (\mu_{(j)} - r^j(t))\Big] \text{ with Explo} = \{T_{\text{init}} + 1, \ldots, T\} \setminus \text{Comm.}
\end{cases}
$$

A **communication step** is defined as a time step where a player is communicating statistics, i.e., using **Send Protocol**. These terms respectively correspond to the regret due to the initialization phase, the communication and the regret of both exploration and exploitation phases.

### A.2.1 Initialization analysis

The initialization regret is obviously bounded by $M(T_0+2K)$ as the initialization phase lasts $T_0+2K$ steps. Lemma 2 provides the probability to reach an orthogonal setting at time $T_0$. If this orthogonal setting is reached, the initialization phase is **successful**. In that case, the players then determine $M$ and a unique internal rank using Pseudocode 5. This is shown by observing that players with external ranks $k$ and $k'$ will exactly collide at round $T_0 + k + k'$.

**Lemma 2.** *After a time $T_0$, all players pull different arms with probability at least $1 - M \exp\left(-\frac{T_0}{K}\right)$.*

*Proof.* As there is at least one arm that is not played by all the other players at each time step, the probability of having no collision at time $t$ for a single player $j$ is lower bounded by $\frac{1}{K}$. It thus holds:

$$\mathbb{P}\left[\forall t \leq T_0, \eta^j(t) = 1\right] \leq \left(1 - \frac{1}{K}\right)^{T_0} \leq \exp\left(-\frac{T_0}{K}\right).$$

For a single player $j$, her probability to encounter only collisions until time $T_0$ is at most $\exp\left(-\frac{T_0}{K}\right)$. The union bound over the $M$ players then yields the desired result. $\quad\square$

### A.2.2 Exploration regret

This section aims at proving Lemma 3, which bounds the exploration regret.

**Lemma 3.** *With probability $1 - \mathcal{O}\left(\frac{K \log(T)}{T} + M \exp\left(-\frac{T_0}{K}\right)\right)$,*

$$R^{\text{explo}} = \mathcal{O}\left(\sum_{k>M} \min\left\{\frac{\log(T)}{\mu_{(M)} - \mu_{(k)}}, \sqrt{T \log(T)}\right\}\right).$$

The proof of Lemma 3 is divided in several auxiliary lemmas. It first relies on the correctness of the estimations before taking the decision to accept or reject any arm.

**Lemma 4.** *For any arm $k$ and positive integer $n$, $\mathbb{P}[\exists p \leq n : |\widetilde{\mu}_k(p) - \mu_k| \geq B_{T_k(p)}] \leq \frac{4n}{T}$.*

*Proof.* For any arm $k$ and positive integer $n$, Hoeffding inequality gives the following, classical inequality in MAB: $\mathbb{P}[\exists p \leq n : |\hat{\mu}_k(p) - \mu_k| \geq \sqrt{\frac{2\log(T)}{T_k(p)}}] \leq \frac{2n}{T}$. It remains to bound the estimation error due to quantization.

Notice that $\sum_{j=1}^M (\widetilde{S}_k^j - \lfloor S_k^j \rfloor)$ is the sum of $M$ independent Bernoulli at each phase $p$. Hoeffding inequality thus also claims that $\mathbb{P}[|\sum_{j=1}^M (\widetilde{S}_k^j(p) - S_k^j(p))| \geq \sqrt{\frac{\log(T)M}{2}}] \leq \frac{2}{T}$. As $T_k(p) \geq M$, it then holds $\mathbb{P}[\exists p \leq n : |\widetilde{\mu}_k^j(p) - \hat{\mu}_k^j(p)| \geq \sqrt{\frac{\log(T)}{2T_k(p)}}] \leq \frac{2n}{T}$. Using the triangle inequality with this bound and the first Hoeffding inequality of the proof yields the final result. $\quad\square$

For both exploration and exploitation phases, we control the number of times an arm is pulled before being accepted or rejected.

**Proposition 1.** *With probability $1 - \mathcal{O}\left(\frac{K \log(T)}{T} + M \exp\left(-\frac{T_0}{K}\right)\right)$, every optimal arm $k$ is accepted after at most $\mathcal{O}\left(\frac{\log(T)}{(\mu_k - \mu_{(M+1)})^2}\right)$ pulls during exploration phases, and every sub-optimal arm $k$ is rejected after at most $\mathcal{O}\left(\frac{\log(T)}{(\mu_{(M)} - \mu_k)^2}\right)$ pulls during exploration phases.*

*Proof.* With probability at least $1 - M \exp\left(-\frac{T_0}{K}\right)$, the initialization is successful, i.e., all players have been assigned different ranks. The remaining of the proof is conditioned on that event.

As there are at most $\log_2(T)$ exploration-communication phases, $|\widetilde{\mu}_k(p) - \mu_k| \leq B_{T_k(p)}$ holds for any arm and phase with probability $1 - \mathcal{O}\left(\frac{K \log(T)}{T}\right)$ thanks to Lemma 4. The remaining of the proof is conditioned on that event.

We first consider an optimal arm $k$. Let $\Delta_k = \mu_k - \mu_{(M+1)}$ be the gap between the arm $k$ and the first sub-optimal arm. We assume $\Delta_k > 0$ here, the case of equality holds considering $\frac{\log(T)}{0} = \infty$. Let $s_k$ be the first integer such that $4B_{s_k} \leq \Delta_k$.

With $T_k(p) = \sum_{l=1}^{p} M_l 2^l$ the number of times an active arm has been pulled after the $p$-th exploration phase, it holds that

$$T(p+1) \leq 3T(p) \qquad \text{as } M_p \text{ is non-increasing.} \tag{2}$$

For some $p \in \mathbb{N}$, $T(p-1) < s_k \leq T(p)$ or the arm $k$ is active at time $T$. In the second case, it is obvious that $k$ is pulled less than $\mathcal{O}(s_k)$ times. Otherwise, the triangle inequality for such a $p$, for any active sub-optimal arm $i$, yields $\widetilde{\mu}_k(p) - B_{T_k(p)} \geq \widetilde{\mu}_i(p) + B_{T_i(p)}$.

So the arm $k$ is accepted after at most $p$ phases. Using the same argument as in [23], it holds $s_k = \mathcal{O}\left(\frac{\log(T)}{(\mu_k - \mu_{(M+1)})^2}\right)$, and also for $T_k(p)$ thanks to Equation (2). Also, $k$ can not be wrongly rejected conditionally on the same event, as it can not be dominated by any sub-optimal arm in term of confidence intervals.

The proof for the sub-optimal case is similar if we denote $\Delta_k = \mu_{(M)} - \mu_k$. $\qquad \square$

In the following, we keep the notation $t_k = \min\left\{\frac{c \log(T)}{\left(\mu_k - \mu_{(M)}\right)^2}, T\right\}$, where $c$ is a universal constant such that with the probability considered in Proposition 1, the number of exploration pulls before accepting/rejecting $k$ is at most $t_k$.

For both exploration and exploitation phases, the decomposition used in the centralized case [3] holds because there is no collision during these two types of phases (conditionally on the success of the initialization phase):

$$R^{\text{explo}} = \sum_{k>M} (\mu_{(M)} - \mu_{(k)})T_{(k)}^{\text{explo}} + \sum_{k\leq M} (\mu_{(k)} - \mu_{(M)})(T^{\text{explo}} - T_{(k)}^{\text{explo}}), \tag{3}$$

where $T^{\text{explo}} = \#\text{Explo}$ and $T_{(k)}^{\text{explo}}$ is the centralized number of time steps where the $k$-th best arm is pulled during exploration or exploitation phases.

**Lemma 5.** *With probability* $1 - \mathcal{O}\left(\frac{K \log(T)}{T} + M \exp\left(-\frac{T_0}{K}\right)\right)$, *the following hold simultaneously:*

    *i) for a sub-optimal arm* $k$, $(\mu_{(M)} - \mu_k)T_k^{\text{explo}} = \mathcal{O}\left(\min\left\{\frac{\log(T)}{\mu_{(M)} - \mu_k}, \sqrt{T \log(T)}\right\}\right)$.

    *ii)* $\sum_{k\leq M} (\mu_{(k)} - \mu_{(M)})(T^{\text{explo}} - T_{(k)}^{\text{explo}}) = \mathcal{O}\left(\sum_{k>M} \min\left\{\frac{\log(T)}{\mu_{(M)} - \mu_{(k)}}, \sqrt{T \log(T)}\right\}\right)$.

*Proof.* i) From Proposition 1, $T_k^{\text{explo}} \leq \mathcal{O}\left(\min\left\{\frac{\log(T)}{(\mu_{(M)} - \mu_k)^2}, T\right\}\right)$ with the considered probability, so $(\mu_{(M)} - \mu_k)T_k^{\text{explo}} = \mathcal{O}\left(\min\left\{\frac{\log(T)}{(\mu_{(M)} - \mu_k)}, (\mu_{(M)} - \mu_k)T\right\}\right)$. The function $\Delta \mapsto \min\left\{\frac{\log(T)}{\Delta}, \Delta T\right\}$ is maximized for $\Delta = \sqrt{\frac{\log(T)}{T}}$ and its maximum is $\sqrt{T \log(T)}$. Thus, the inequality $\min\left\{\frac{\log(T)}{\Delta}, \Delta T\right\} \leq \min\left\{\frac{\log(T)}{\Delta}, \sqrt{T \log(T)}\right\}$ always holds for $\Delta \geq 0$ and yields the first point.

ii) We (re)define the following: $\hat{t}_k$ the number of exploratory pulls before accepting/rejecting the arm $k$, $M_l$ the number of active player during the $l$-th exploration phase, $T(p) = \sum\limits_{l=1}^{p} 2^l M_l$ and $N$ the total number of exploration phases.

$T(p)$ describes the total number of exploration pulls processed at the end of the $p$-th exploration phase on every active arm for $p < N$. Since the $N$-th phase may remain uncompleted, $T(N)$ is then greater that the number of exploration pulls at the end of the $N$-th phase.

With probability $1 - \mathcal{O}\left(\frac{K \log(T)}{T} + M \exp\left(-\frac{T_0}{K}\right)\right)$, the initialization is successful, any arm is correctly accepted or rejected and $\hat{t}_k \leq t_k$ for all $k$. The remaining of the proof is conditioned on that event. We now decompose the proof in two main parts given by Lemmas 6 and 7 proven below.

**Lemma 6.** *Conditionally on the success of the initialization phase and on correct estimations of all arms:*

$$\sum_{k \leq M} (\mu_{(k)} - \mu_{(M)})(T^{\text{explo}} - T^{\text{explo}}_{(k)}) \leq \sum_{j > M} \sum_{k \leq M} \sum_{p=1}^{N} 2^p (\mu_{(k)} - \mu_{(M)}) \mathbb{1}_{\min(\hat{t}_{(j)}, \hat{t}_{(k)}) > T(p-1)}.$$

**Lemma 7.** *Conditionally on the success of the initialization phase and on correct estimations of all arms:*

$$\sum_{k \leq M} \sum_{p=1}^{N} 2^p (\mu_{(k)} - \mu_{(M)}) \mathbb{1}_{\min(\hat{t}_{(j)}, \hat{t}_{(k)}) > T(p-1)} \leq \mathcal{O}\left(\min\left\{\frac{\log(T)}{\mu_{(M)} - \mu_{(j)}}, \sqrt{T \log(T)}\right\}\right).$$

These two lemmas directly yield the second point in Lemma 5. $\qquad\square$

*Proof of Lemma 6.* Let us consider an optimal arm $k$. During the $p$-th exploration phase, there are two possibilities:

- either $k$ has already been accepted, i.e., $\hat{t}_k \leq T(p-1)$. Then the arm $k$ is pulled the whole phase, i.e., $K_p 2^p$ times.

- Or $k$ is still active. Then it is pulled $2^p$ times by each active player, i.e., it is pulled $M_p 2^p$ times in total. This means that it is not pulled $(K_p - M_p)2^p$ times.

From these two points, it holds that $T^{\text{explo}}_k \geq T^{\text{explo}} - \sum\limits_{p=1}^{N} 2^p (K_p - M_p) \mathbb{1}_{\hat{t}_k > T(p-1)}$.

Notice that $K_p - M_p$ is the number of active sub-optimal arms. By definition, $K_p - M_p = \sum\limits_{j > M} \mathbb{1}_{\hat{t}_{(j)} > T(p-1)}$. We thus get that $T^{\text{explo}}_k \geq T^{\text{explo}} - \sum\limits_{j > M} \sum\limits_{p=1}^{N} 2^p \mathbb{1}_{\min(\hat{t}_{(j)}, \hat{t}_k) > T(p-1)}$.

The double sum actually is the number of times a sub-optimal arm is pulled instead of $k$. This yields the result when summing over all optimal arms $k$. $\qquad\square$

*Proof of Lemma 7.* Let us define $A_j = \sum\limits_{k \leq M} \sum\limits_{p=1}^{N} 2^p (\mu_{(k)} - \mu_{(M)}) \mathbb{1}_{\min(\hat{t}_j, \hat{t}_{(k)}) > T(p-1)}$ the cost associated to the sub-optimal arm $j$. Lemma 7 upper bounds $A_j$ for any sub-optimal arm $j$.

Recall that $t_{(k)} = \min\left(\frac{c \log(T)}{\left(\mu_{(k)} - \mu_{(M)}\right)^2}, T\right)$ for a universal constant $c$. The proof is conditioned on the event $\hat{t}_{(k)} \leq t_{(k)}$, so that if we define $\Delta(p) = \sqrt{\frac{c \log(T)}{T(p-1)}}$, the inequality $\hat{t}_{(k)} > T(p-1)$ implies $\mu_{(k)} - \mu_{(M)} < \Delta(p)$. We also write $N^j$ the first integer such that $\hat{t}_j \leq T(N^j)$. It follows:

$$A_j \leq \sum_{k \leq M} \sum_{p=1}^{N^j} 2^p \Delta(p) \mathbb{1}_{\hat{t}_{(k)} > T(p-1)}$$

$$\leq \sum_{p=1}^{N^j} \Delta(p)\left(T(p) - T(p-1)\right) \qquad\qquad \text{as } \sum_{k \leq M} \mathbb{1}_{\hat{t}(k) > T(p-1)} = M_p.$$

$$= c\log(T)\sum_{p=1}^{N^j} \Delta(p)\left(\frac{1}{\Delta(p+1)} + \frac{1}{\Delta(p)}\right)\left(\frac{1}{\Delta(p+1)} - \frac{1}{\Delta(p)}\right)$$

$$\leq (1+\sqrt{3})c\log(T)\sum_{p=1}^{N^j}\left(\frac{1}{\Delta(p+1)} - \frac{1}{\Delta(p)}\right) \qquad\qquad \text{thanks to Equation (2)}.$$

$$\leq (1+\sqrt{3})c\log(T)\frac{1}{\Delta(N^j+1)} \qquad\qquad \text{by convention, } \frac{1}{\Delta(1)} = 0.$$

By definition of $N^j$, we have $t_j \geq T(N^j - 1)$. Thus, $\Delta(N^j) \geq \sqrt{\frac{c\log(T)}{t_j}}$ and Equation (2) gives $\Delta(N^j + 1) \geq \sqrt{\frac{c\log(T)}{3t_j}}$. It then holds $A_j \leq (3+\sqrt{3})\sqrt{c\,t_j\log(T)}$. The result follows since $t_j = \mathcal{O}\left(\min\left\{\frac{\log(T)}{(\mu_{(M)} - \mu_j)^2}, T\right\}\right)$. $\qquad\square$

Using the two points of Lemma 5, along with Equation (3), yields Lemma 3.

### A.2.3  Communication cost

We now focus on the $R^{\mathrm{comm}}$ term in Equation (1). Lemma 8 states it is negligible compared to $\log(T)$ and has a significant impact on the regret only for small values of $T$.

**Lemma 8.** *With probability* $1 - \mathcal{O}\left(\frac{K\log(T)}{T} + M\exp\left(-\frac{T_0}{K}\right)\right)$, *the following holds:*

$$R^{\mathrm{comm}} = \mathcal{O}\left(KM^3\log^2\left(\min\left\{\frac{\log(T)}{(\mu_{(M)} - \mu_{(M+1)})^2}, T\right\}\right)\right).$$

*Proof.* As explained in Section 2.2.3, the length of the communication phase $p \in [N]$ is at most $KM^2(p+1)$, where $N$ is the number of exploration phases. The cost of communication is then smaller than $KM^3\sum_{p=1}^N (p+1) \leq \mathcal{O}\left(KM^3N^2\right)$. Proposition 1 in Appendix A.2.2, claims with the considered probability that $N$ is at most $\mathcal{O}\left(\log\left(\min\left\{\frac{\log(T)}{(\mu_{(M)} - \mu_{(M+1)})^2}, T\right\}\right)\right)$, which yields Lemma 8. $\qquad\square$

### A.2.4  Total regret

The choice $T_0 = \lceil K\log(T)\rceil$ along with Lemmas 2, 3 and 8 claim that a bad event occurs with probability at most $\mathcal{O}\left(\frac{K\log(T)}{T} + \frac{M}{T}\right)$. The average regret due to bad events is thus upper bounded by $\mathcal{O}(KM\log(T))$. Using these lemmas along with Equation (1) finally yields the bound in Theorem 1.

### A.3  Experiments

We compare in Figure 1 the empirical performances of SIC-MMAB with the MCTopM algorithm[8] on generated data[4]. We also compared with the MusicalChairs algorithm [26], but its performance was irrelevant and out of scale. This is mainly due to its scaling with $1/\Delta^2$, besides presenting large constant terms in its regret. Also, its main advantage comes from its scaling with $M$, which is here small for computational reasons. All the considered regret values are averaged over 200 runs. The experiments are run with Bernoulli distributions. Thus, there is no need to quantize the sent statistics and a tighter confidence bound $B_s = \sqrt{\frac{2\log(T)}{s}}$ is used.

Figure 1a represents the evolution of the regret for both algorithms with the following problem parameters: $K = 9$, $M = 6$, $T = 5 \times 10^5$. The means of the arms are linearly distributed between

0.9 and 0.89, so the gap between two consecutive arms is $1.25 \times 10^{-3}$. The switches between exploration and communication phases for SIC-MMAB are easily observable. A larger horizon (near 40 times larger) is required for SIC-MMAB to converge to a constant regret, but this alternation between the phases could not be visible for such a value of $T$.

Figure 1b represents the evolution of the final regret as a function of the gap $\Delta$ between two consecutive arms in a logarithmic scale. The problem parameters $K$, $M$ and $T$ are the same. Although MCTopM seems to provide better results with larger values of $\Delta$, SIC-MMAB seems to have a smaller dependency in $1/\Delta$. This confirms the theoretical results claiming that MCTopM scales with $\Delta^{-2}$ while SIC-MMAB scales with $\Delta^{-1}$. This can be observed on the left part of Figure 1b where the slope for MCTopM is approximately twice as large as for SIC-MMAB. Also, a different behavior of the regret appears for very low values of $\Delta$ which is certainly due to the fact that the regret only depends on $T$ for extremely small values of $\Delta$ (minmax regret).

(a) Evolution of regret over time.  (b) Final regret as a function of $\frac{1}{\Delta}$.

Figure 1: Performance comparison between SIC-MMAB and MCTopM algorithms.

## B  Complementary material for Section 3

### B.1  DYN-MMAB description

This section thoroughly describes DYN-MMAB algorithm. Its pseudocode is given in Algorithm 2 below. We first describe the rules explaining when a player adds an arm to `Occupied` or `Preferences`.

An arm $k$ is added to `Occupied` (it may already be in `Preferences`) if only 0 rewards have been observed during a whole block of $L_k^j$ pulls on arm $k$ for player $j$. Such a block ends when $L_k^j$ observations have been gathered on arm $k$ and a new block is then restarted. $L_k^j$ is an estimation of the required number of successive 0 to observe before considering an arm as occupied with high probability. Its value at stage $t + \tau_j$, $L_k^j(t)$, is thus constantly updated using the current estimation of a lower bound of $\mu_k$:

$$L_k^j(t+1) \leftarrow \min\left( \frac{2e \log(T^j)}{\left( \hat{r}_k^j(t+1) - B^j(t+1) \right)^+}, L_k^j(t) \right) \quad \text{and } L_k^j(0) = +\infty, \quad (4)$$

where $\hat{r}_k^j(t)$ is the empirical mean reward on the arm $k$ at stage $t + \tau_j$, $B^j(t) = 2\sqrt{\frac{6\,K \log(T^j)}{t}}$, $x^+ = \max(x, 0)$ and $\frac{2e \log(T^j)}{0} = +\infty$. This rule is described at lines 12-15 in Algorithm 2.

An active arm $k$ is added to `Preferences` (at last position) if it is better than any other active arm, in term of confidence interval. This rule is described at lines 16-18 in Algorithm 2.

Another rule needs to be added to handle the possible case of an arm in `Preferences` already exploited by another player. As soon as an arm $k$ in `Preferences` becomes worse (in terms of

---

**Algorithm 2** DYN-MMAB algorithm

---

**Input:** $T^j$ (personal horizon)

1: $p \leftarrow 1$, Fixed $\leftarrow -1$ and initialize Preferences, Occupied as empty lists
2: $\mathbf{T}, \mathbf{T}^{\text{temp}}, \mathbf{S}, \mathbf{S}^{\text{temp}} \leftarrow \text{Zeros}(K)$ and define $\mathbf{L}$ as a vector of $K$ elements equal to $\infty$
3: $r_{\inf}[k] \leftarrow 0$ and $r_{\sup}[k] \leftarrow 1$ for every arm $k$       *# Initialize the confidence intervals*
4: **Exploration Phase:**              *# $B^j(t) = 2\sqrt{\frac{6\,K\log(T^j)}{t}}$ here*
5: **while** Fixed $= -1$ **do**
6:     Pull $k \sim \mathcal{U}([K])$; $T^{\text{temp}}[k] \leftarrow T^{\text{temp}}[k] + 1$ and $T[k] \leftarrow T[k] + 1$
7:     $S^{\text{temp}}[k] \leftarrow S^{\text{temp}}[k] + r_k(t)$ and $S[k] \leftarrow S[k] + r_k(t)$
8:     For all arms $i$, $r_{\inf}[i] \leftarrow \left(\frac{S[i]}{T[i]} - B^j(t)\right)^+$ and $r_{\sup}[i] \leftarrow \min\left(\frac{S[i]}{T[i]} + B^j(t), 1\right)$
9:     $L[k] \leftarrow \min\left(\frac{2e\log(T^j)}{r_{\inf}[k]},\ L[k]\right)$
10:     **if** $k = $ Preferences$[p]$ and $r_k(t) > 0$ **then** Fixed $\leftarrow k$ **end if**     *# no collision*
                                                                         *on the arm to exploit*
11:     **if** Preferences$[p] \in$ Occupied **then** $p \leftarrow p + 1$ **end if**     *# exploited by another player*
12:     **if** $T^{\text{temp}}[k] \geq L[k]$ **then**                   *# end of sliding window*
13:       **if** $S^{\text{temp}}[k] = 0$ **then** Add $k$ to Occupied **end if**     *# estimate that $k$ is occupied*
14:       Reset $S^{\text{temp}}[k], T^{\text{temp}}[k] \leftarrow 0$
15:     **end if**
16:     **if** for some active arm $i$ and all other active arms $l$, $r_{\inf}[i] > r_{\sup}[l]$ **then**
17:       Add $i$ to Preferences (last position)     *# $i$ is better than any other active arm*
18:     **end if**
19:     **if** there is some $l$ not in Preferences$[1:p]$, such that $r_{\inf}[l] > r_{\sup}[$Preferences$[p]]$
      **then** add Preferences$[p]$ to Occupied
20:     **end if**                *# the mean of the available best arm has significantly dropped*
21: **end while**

22: **Exploitation Phase:** Pull Fixed until $T^j$

---

confidence intervals) than an active arm or an arm with a higher index in Preferences, then $k$ is added to Occupied. This rule is described at lines 19-20 in Algorithm 2.

Following these rules, as soon as there is an arm in Preferences, player $j$ tries to occupy the $p$-th arm in Preferences (starting with $p = 1$), yet she still continues to explore. As soon as she encounters a positive reward on it, she occupies it and starts the exploitation phase. If she does not end up occupying an optimal arm, this arm will be added to Occupied at some point. The player then increments $p$ and tries to occupy the next available best arm. This point is described at lines 10-11 in Algorithm 2. Notice that Preferences can have more than $p$ elements, but the player must not exploit the $q$-th element of Preferences with $q > p$ yet as it can lead the player in exploiting a sub-optimal arm.

### B.2 Theoretical analysis

#### B.2.1 Auxiliary lemmas

This section is devoted to the proof of Theorem 2. It first proves the first point of Lemma 1.

*Proof of Lemma 1.1.* We first introduce $Z_t := X_k(t+\tau_j)(1-\eta_k(t+\tau_j))\mathbb{1}_{\pi^j(t+\tau_j)=k}$ and $p_t := \mathbb{E}[Z_t]$. Notice that $p_t \leq \frac{1}{K}$ because $\mathbb{1}_{\pi^j(t+\tau_j)=k}$ is a Bernoulli of parameter $\frac{1}{K}$ in the exploration phase. Chernoff bound states that:

$$\mathbb{P}\Big[\sum_{t'=1}^{t}(Z_{t'} - \mathbb{E}[Z_{t'}]) \geq t\delta\Big] \leq \min_{\lambda > 0} e^{-\lambda t\delta}\, \mathbb{E}\Big[\prod_{t'=1}^{t} e^{\lambda(Z_{t'} - \mathbb{E}[Z_{t'}])}\Big].$$

By convexity, $e^{\lambda z} \leq 1 + z(e^\lambda - 1)$ for $z \in [0,1]$. It thus holds:

$$\mathbb{E}\Big[e^{\lambda(Z_t - \mathbb{E}[Z_t])}\Big] \leq e^{-\lambda p_t}\big(1 + p_t(e^\lambda - 1)\big) \leq e^{-\lambda p_t} e^{p_t(e^\lambda - 1)} \qquad \text{as } 1 + x \leq e^x.$$

$$\leq e^{p_t(e^\lambda - 1 - \lambda)} \leq e^{\frac{e^\lambda - 1 - \lambda}{K}} \qquad\qquad \text{as } p_t \leq \frac{1}{K} \text{ and } e^\lambda - 1 - \lambda \geq 0.$$

It can then be deduced:

$$\mathbb{P}\Big[ \sum_{t'=1}^{t} (Z_{t'} - \mathbb{E}[Z_{t'}]) \geq t\delta \Big] \leq \min_{\lambda > 0} e^{-\lambda t\delta} e^{t \frac{e^\lambda - 1 - \lambda}{K}}. \qquad\qquad \text{For } \lambda = \log(1 + K\delta):$$

$$\leq \exp\left(-\frac{t}{K} h(K\delta)\right) \qquad \text{with } h(u) = (1 + u)\log(1 + u) - u.$$

Similarly, we show for the negative error: $\mathbb{P}\Big[ \sum\limits_{t'=1}^{t} (Z_{t'} - \mathbb{E}[Z_{t'}]) \leq -t\delta \Big] \leq \exp\left(-\frac{t}{K} h(-K\delta)\right)$.

Either $t \leq \frac{16}{3} K \log(T^j)$ and the desired inequality holds almost surely, or $K\delta < 1$ with $\delta = \sqrt{\frac{16 \log(T^j)}{3tK}}$. As $h(x) \geq \frac{3x^2}{8}$ for $|x| < 1$, it then holds

$$\mathbb{P}\Big[ \Big| \sum_{t'=1}^{t} (Z_{t'} - \mathbb{E}[Z_{t'}]) \Big| \geq t\delta \Big] \leq 2 e^{-\frac{3t(K\delta)^2}{8K}} \qquad \text{and after multiplication with } \frac{K}{t}:$$

$$\mathbb{P}\left[ \Big| \frac{K}{t} \sum_{t'=1+\tau_j}^{t+\tau_j} X_k(t')(1 - \eta_k(t')) \mathbb{1}_{\pi^j(t')=k} - \gamma_j(t)\mu_k \Big| \geq \sqrt{\frac{16K \log(T^j)}{3t}} \right] \leq \frac{2}{(T^j)^2}. \qquad (5)$$

Chernoff bound also provides a confidence interval on the number of pulls on a single arm:

$$\mathbb{P}\left[ \Big| T_k^j(t) - \frac{t}{K} \Big| \geq \sqrt{\frac{6t \log(T^j)}{K}} \right] \leq \frac{2}{(T^j)^2}. \qquad (6)$$

From Equation (6), it can be directly deduced that $\mathbb{P}\Big[ |\frac{K T_k^j(t)}{t} - 1| \geq \sqrt{\frac{6K \log(T^j)}{t}} \Big] \leq \frac{2}{(T^j)^2}$. As $\hat{r}_k^j(t) \leq 1$,

$$\mathbb{P}\left[ \Big| \frac{K T_k^j(t)}{t} \hat{r}_k^j(t) - \hat{r}_k^j(t) \Big| \geq \sqrt{\frac{6K \log(T^j)}{t}} \right] \leq \frac{2}{(T^j)^2}. \qquad (7)$$

As $\frac{K T_k^j(t)}{t} \hat{r}_k^j(t) = \frac{K}{t} \sum\limits_{t'=1+\tau_j}^{t+\tau_j} X_k(t')(1 - \eta_k(t')) \mathbb{1}_{\pi^j(t')=k}$, using the triangle inequality with Equations (5) and (7) finally yields $\mathbb{P}\Big[ |\hat{r}_k^j(t) - \gamma^j(t)\mu_k| \geq 2\sqrt{\frac{6\,K \log(T^j)}{t}} \Big] \leq \frac{4}{(T^j)^2}$. $\qquad\qquad \square$

The second point of Lemma 1 is proved below.

*Proof of Lemma 1.2.* The previous point gives that with probability $1 - \mathcal{O}\left(\frac{K}{T^j}\right)$, player $j$ correctly estimated all the free arms until stage $T$. The remaining of the proof is conditioned on this event. We also assume that $t^0$ is the first stage where $k$ is occupied for the proof. The general result claimed in Lemma 1 directly follows.

When $t^0$ is small, the second case will happen, i.e., the number of pulls on the arm $k$ is small and its average reward can quickly drop to 0. When $t^0$ is large, $\gamma_j(t)\mu_k$ is tightly estimated so that $L_k^j$ is small. Then, the first case will happen, i.e., the arm $k$ will be quickly detected as occupied.

a) We first assume $t^0 \leq 12K \log(T^j)$. The empirical reward after $T_k^j(t) \geq T_k^j(t^0)$ pulls is $\hat{r}_k^j(t) = \frac{\hat{r}_k^j(t^0) T_k^j(t^0)}{T_k^j(t)}$, because all pulls after the stage $t^0 + \tau_j$ will return 0 rewards. However, using Chernoff bound as in Equation (6), it appears that if $t^0 \leq 12K \log(T^j)$ then $T_k^j(t^0) \leq 18 \log(T^j)$ with probability $1 - \mathcal{O}\left(\frac{1}{T^j}\right)$, so $\hat{r}_k^j(t) \leq \frac{18 \log(T^j)}{T_k^j(t)}$.

Conditionally on the correct estimations of the arms, there is at least an unoccupied arm $i$ with $\mu_i \leq \mu_k$. Therefore with $t_i = \frac{72Ke\log(T^j)}{\mu_i^2}$, as $t_i \geq 12K\log(T^j)$, Chernoff bound guarantees that the following holds, with probability at least $1 - \frac{2}{T^j}$,

$$\frac{3t_i}{2K} \geq T_k^j(t_i) \geq \frac{t_i}{2K} = \frac{36e\log(T^j)}{\mu_i^2}. \tag{8}$$

This gives that $\hat{r}_k^j(t_i) \leq \frac{\mu_i}{2e}$. After stage $\tau_j + \frac{d'K\log(T^j)}{\mu_i^2}$, where $d'$ is some universal constant, the error bounds of both arms are upper bounded by $\frac{\mu_i}{8e}$. The confidence intervals would then be disjoint for the arms $k$ and $i$. So $k$ will be detected as worse than $i$ after a time at most $\mathcal{O}\left(\frac{K\log(T)}{\mu_i^2}\right)$ as $T^j \leq T$.

b) We now assume that $12K\log(T^j) \leq t^0 \leq \frac{24\lambda K\log(T^j)}{\mu_k^2}$ with $\lambda = 16e^2$. It still holds $\hat{r}_k^j(t) = \frac{\hat{r}_k^j(t^0)T_k^j(t^0)}{T_k^j(t)}$. Correct estimations of the free arms are assumed in this proof, so in particular

$$\hat{r}_k^j(t) \leq \frac{(\mu_k + B^j(t^0))T_k^j(t^0)}{T_k^j(t)}. \tag{9}$$

As in Equation (8), it holds that $T_k^j(t^0) \leq \frac{3t^0}{2K}$ with probability $1 - \mathcal{O}\left(\frac{1}{T^j}\right)$ and thus $B^j(t^0) \leq 6\sqrt{\frac{\log(T^j)}{T_k^j(t^0)}}$. Also, $T_k^j(t) \geq \frac{d\log(T^j)}{2\mu_i\mu_k}$ for $t = d\frac{K\log(T^j)}{\mu_i^2}$. Equation (9) then becomes

$$\hat{r}_k^j(t) \leq \frac{\mu_k T_k^j(t^0)}{T_k^j(t)} + \frac{B^j(t^0)T_k^j(t^0)}{T_k^j(t)} \leq \frac{36\lambda}{d}\mu_i + \frac{6\sqrt{T_k^j(t^0)\log(T^j)}}{T_k^j(t)} \leq \left(\frac{36\lambda}{d} + \frac{72\sqrt{\lambda}}{d}\right)\mu_i.$$

Thus, for a well chosen $d$, the empirical reward verifies $\hat{r}_k^j(t) \leq \frac{\mu_i}{2e}$. We then conclude as for the first case that the arm $k$ would be detected as worse than the free arm $i$ after a time $\mathcal{O}\left(\frac{K\log(T)}{\mu_i^2}\right)$.

c) The last case corresponds to $t^0 > \frac{24\lambda K\log(T^j)}{\mu_k^2}$. It then holds $B^j(t^0) \leq \frac{\mu_k}{\sqrt{\lambda}} = \frac{\mu_k}{4e}$.

By definition, $L_k^j \leq \frac{2e\log(T^j)}{\hat{r}_k^j - B^j(t)}$. Conditionally on the correct estimation of the free arms, it holds that $\gamma_j(t)\mu_k - 2B^j(t) \leq \hat{r}_k^j - B^j(t) \leq \mu_k$. So with the choice of $L_k^j$ described by Equation (4), as long as $k$ is free,

$$\frac{2e\log(T^j)}{\mu_k} \leq L_k^j \leq \frac{2e\log(T^j)}{\gamma_j(t)\mu_k - 2B^j(t)} \leq \frac{2e^2\log(T^j)}{\mu_k - 2eB^j(t)}. \tag{10}$$

As $B^j(t^0) \leq \frac{\mu_k}{4e}$, it holds that $L_k^j(t^0) \leq \frac{4e^2\log(T^j)}{\mu_k}$. Since $L_k^j$ is non-increasing by definition, this actually always holds for any $t$ larger than $t^0$.

From that point, Equation (8) gives that with probability $1 - \mathcal{O}\left(\frac{1}{T^j}\right)$, the arm $k$ will be pulled at least $2L_k^j$ times between stage $t^0 + 1$ and $t^0 + 24KL_k^j$ with probability $1 - \mathcal{O}\left(\frac{1}{T^j}\right)$. Thus, a whole block of $L_k^j$ pulls receiving only 0 rewards on $k$ happens before stage $t^0 + 24KL_k^j$.

The arm $k$ is then detected as occupied after a time $\mathcal{O}\left(\frac{K\log(T^j)}{\mu_k}\right)$ from $t^0$, leading to the result. $\quad\square$

**Lemma 9.** *At any stage, no free arm $k$ is falsely detected as occupied by player $j$ with probability $1 - \mathcal{O}\left(\frac{K}{T^j}\right)$.*

*Proof.* As shown above, with probability $1 - \mathcal{O}\left(\frac{K}{T^j}\right)$, player $j$ correctly estimated the average rewards of all the free arms until stage $T$. The remaining of the proof is conditioned on that event. As long as $k$ is free, it can not become dominated by some arm that was not added to `Preferences` before $k$, so it can not be added to `Occupied` from the rule given at lines 19-20 in Algorithm 2.

For the rule of lines 12-15, Equation (10) gives that

$$L_k^j(t') \geq \frac{2e \log(T^j)}{\mu_k} \qquad \text{at each stage } t' \leq t. \tag{11}$$

As in Appendix A.2.1, the probability of detecting $L$ successive 0 rewards on a free arm $k$ is then smaller than $\left(1 - \frac{\mu_k}{e}\right)^L \leq \exp\left(-\frac{L\mu_k}{e}\right)$.

Using this along with Equation (11) yields that with probability $1 - \mathcal{O}\left(\frac{1}{(T^j)^2}\right)$, at least one positive reward will be observed on arm $k$ in a single block. The union bound over all blocks yields the result. $\square$

Finally, Lemma 10 yields that, after some time, any player starts exploiting an arm while all the better arms are already occupied by other players.

**Lemma 10.** *We denote $\bar{\Delta}_{(k)} = \min_{i=1,\ldots,k} (\mu_{(i)} - \mu_{(i+1)})$. With probability $1 - \mathcal{O}\left(\frac{K}{T^j}\right)$, it holds that for a single player $j$, there exists $k_j$ such that after a stage at most $\bar{t}_{k_j} + \tau_j$, she is exploiting the $k_j$-th best arm and all the better arms are also exploited by other players, where*
$$\bar{t}_{k_j} = \mathcal{O}\left(\frac{K \log(T)}{\bar{\Delta}_{(k_j)}^2} + k_j \frac{K \log(T)}{\mu_{(k_j)}}\right).$$

*Proof.* Player $j$ correctly estimates all the arms until stage $T$, with probability $1 - \mathcal{O}\left(\frac{K}{T^j}\right)$. The remaining of the proof is conditioned on that event. We define $\bar{t}_i = \frac{cK \log(T^j)}{\bar{\Delta}_{(i)}^2} + i\frac{cK \log(T^j)}{\mu_{(i)}}$ for some universal constant $c$ and $k_j$ (random variable) defined as

$$k_j = \min\left\{i \in [K] \mid i\text{-th best arm not exploited by another player at stage } \bar{t}_i + \tau_j\right\}. \tag{12}$$

$k_j^*$ ($k_j$-th best arm) is the best arm not exploited by another player (than player $j$) after the stage $\bar{t}_{k_j} + \tau_j$. The considered set is not empty as $M \leq K$.

Lemma 9 gives that with probability $1 - \mathcal{O}\left(\frac{K}{T^j}\right)$, $k_j^*$ is not falsely detected as occupied until stage $T$. All arms below $k_j^*$ will be detected as worse than $k_j^*$ after a time $\frac{dK \log(T^j)}{\bar{\Delta}_{(k_j)}^2}$ for some universal constant $d$.

By definition of $k_j$, any arm $i^*$ better than $k_j^*$ is already occupied at stage $\bar{t}_i + \tau_j$. Lemma 1, gives that with probability $1 - \mathcal{O}\left(\frac{1}{T^j}\right)$, either $i^*$ is detected as occupied after stage $\bar{t}_i + \tau_j + \frac{d'K \log(T^j)}{\mu_{(i)}}$ or dominated by $k_j^*$ after stage $\frac{d_2 K \log(T^j)}{\bar{\Delta}_{(k_j)}^2} + \tau_j$ for some universal constants $d'$ and $d_2$.

Thus the player detects the arm $k_j^*$ as optimal and starts trying to occupy $k_j^*$ at a stage at most $\tilde{t} = \max\left(\bar{t}_{k_j - 1} + \frac{d'K \log(T^j)}{\mu_{(k_j)}}, \max(d, d_2)\frac{K \log(T^j)}{\bar{\Delta}_{(k_j)}^2}\right) + \tau_j$ with probability $1 - \mathcal{O}\left(\frac{K}{T^j}\right)$ (where $\bar{t}_0 = 0$).

Using similar arguments as for Lemma 9, player $j$ will observe a positive reward on $k_j^*$ with probability $1 - \mathcal{O}\left(\frac{1}{T^j}\right)$ after a stage at most $\tilde{t} + \frac{d_2'K \log(T^j)}{\mu_{(k_j)}}$ for some constant $d_2'$, if $k_j$ is still free at this stage. With the choice $c = \max(d, d_2, d' + d_2')$, this stage is smaller than $\bar{t}_{k_j}$ and $k_j^*$ is then still free. Thus, player $j$ will start exploiting $k_j^*$ after stage at most $\bar{t}_{k_j}$ with the considered probability. $\square$

### B.2.2 Regret in dynamic setting

*Proof of Theorem 2.* Lemma 10 states that a player only needs an exploration time bounded as $\mathcal{O}\left(\frac{K\log(T)}{\tilde{\Delta}_{(k)}^2} + k\frac{K\log(T)}{\mu_{(k)}}\right)$ before starting exploiting, with high probability. Furthermore, the better arms are already exploited when she does so. Thus, the exploited arms are the top-$M$ arms. The regret is then upper bounded by twice the sum of exploration times (and the low probability events of wrong estimations), as a collision between players can only happen with at most one player in her exploitation phase.

The regret incurred by low probability events mentioned in Lemma 10 is in $\mathcal{O}(KM^2)$ and is thus dominated by the exploration regret. □

## C   No Sensing: communication through synchronization

This section focuses on the static No Sensing model. First of all, we claim that a communication protocol similar to the one of SIC-MMAB can be devised here, under a mild extra assumption: a lower bound $\mu_{\min}$ of the average rewards $\mu_k$ is known[5]. Indeed, in the Collision Sensing model, a bit is sent through a single collision. Without sensing, it can be done with probability $1 - \frac{1}{T}$ using $\frac{\log(T)}{\mu_{\min}}$ time steps. This adds a multiplicative factor of $\frac{\log(T)}{\mu_{\min}}$ to the communication regret[6], which would then dominate the new initialization regret. So, SIC-MMAB can be easily adapted for the No Sensing model into the ADAPTED SIC-MMAB algorithm with a regret scaling as

$$\mathcal{O}\left(\sum_{k>M}\frac{\log(T)}{\mu_{(M)}-\mu_{(k)}} + \frac{KM^3\log(T)}{\mu_{\min}}\log^2\left(\frac{\log(T)}{(\mu_{(M)}-\mu_{(M+1)})^2}\right)\right). \tag{13}$$

The exploration regret is still similar to the centralized algorithm, but the communication cost is no longer sub-logarithmic. In this section, we introduce an alternative algorithm for the No sensing setting. It also relies on a communication protocol, but with more limited information, which thus incurs a much better dependency in $M$ as well as a logarithmic regret.

In the No Sensing setting, the SELFISH strategy, where all players follow independently UCB seems to perform well (on generated data) but appears to incur a linear regret with some constant probability [8]. In Appendix D, the discussion about the SELFISH algorithm is extended and some reasons of its failure are explained, using algebraic arguments (Lindemann-Weierstrass Theorem).

### C.1   Adapted communication protocol

The algorithm SIC-MMAB2 is formally described in Appendix C.2. It relies on several subroutines that are detailed in the next section. Similarly to SIC-MMAB, it starts with an initialization phase to estimate $M$. It then alternates between exploration and communication phases, but the goal of the communication phases is here to communicate to other players that an arm is optimal or sub-optimal (instead of transmitting statistics). This allows to share common sets of active arms and players. Protocols to declare such arms and to detect declarations from other players are detailed in Appendix C.2.3. The algorithm then ends with an exploitation phase.

An additional assumption is required for SIC-MMAB2 and is quite similar to an assumption made by Lugosi and Mehrabian [21] for the No Sensing model.

**Assumption 3.** *A lower bound of $\mu_{(K)}$ is known to all players:* $0 < \mu_{\min} \leq \min_i \mu_i$.

The regret incurred by SIC-MMAB2 is given by Theorem 3. Its proof is given in Appendix C.3.

**Theorem 3.** *With the choice $T_c = \lceil \frac{\log(T)}{\mu_{\min}} \rceil$ for the initialization,* SIC-MMAB2 *has a regret scaling as*

$$\mathbb{E}[R_T] = \mathcal{O}\left( \sum_{k>M} \min\left\{ \frac{M\log(T)}{\mu_{(M)} - \mu_{(k)}}, \sqrt{MT\log(T)} \right\} + \frac{MK^2}{\mu_{\min}}\log(T) \right).$$

## C.2  Algorithm description

SIC-MMAB2 algorithm is described in this section. We use the same definitions for $M_p$ and $K_p$ as in Section 2.

### C.2.1  Initialization phase

The objective of the initialization phase is to estimate $M$. First, each player follows the Musical Chairs algorithm for a time $KT_c$ with $T_c := \lceil \frac{\log(T)}{\mu_{\min}} \rceil$. The algorithm in the No Sensing setting is given by Pseudocode 4, Appendix A.1. The second protocol of the initialization is then the same as for SIC-MMAB, but instead of a single time step, a number of $T_c$ time steps is needed to correctly transmit a bit with probability $1 - \frac{1}{T}$. The detailed protocol is given by Pseudocode 6.

---

**Estimate_M_NoSensing Protocol**

---

    **Input:** $k$ (external rank), $T_c$ (time to send a bit)
    **Output:** $M$ (estimated number of players)
1: Initialize $M \leftarrow 1$ and $\pi \leftarrow k$
2: **for** $n = 1, \dots, 2K$ **do**
3:     Initialize $r \leftarrow 0$
4:     **if** $n \geq 2k$ **then** $\pi \leftarrow \pi + 1 \pmod{K}$ **end if**           *# sequential hopping*
5:     **for** $T_c$ time steps **do** Pull $\pi$ and update $r \leftarrow r + r_\pi(t)$ **end for**
6:     **if** $r = 0$ **then** $M \leftarrow M + 1$ **end if**                *# increases if $T_c$ collisions*
7: **end for**
8: **return** $M$

---

Pseudocode 6: estimate $M$ for the No Sensing setting.

### C.2.2  Exploration phases

Each exploration phase is split into two parts. During the first one, each player fixes to an arm following Musical Chairs procedure. After this procedure, players are in an orthogonal setting and can thus start the second part, where they hop sequentially and explore the active arms without any collision. The decisions for accepting/rejecting arms are still based on the exploration pulls as in SIC-MMAB. The differences with the exploration of SIC-MMAB are the following:

1. statistics are not shared among players; this induces an additional $M$ factor in the regret.

2. A Musical Chairs procedure is added at the beginning of a new exploration phase, if there was at least one declaration or fixation block in the previous communication phase. This procedure is needed to reach an orthogonal setting before the sequential hopping. Figures 2 and 3 below illustrate when such a procedure is added. It corresponds to lines 8-11 in Algorithm 3.

Figure 2: Alternating between fixation, exploration and declaration blocks. Case where a player declares sub-optimal arms and tries to occupy (without success) optimal arms.

Figure 3: Alternating between fixation, exploration and declaration blocks. Case with no declaration. In that case, the single declaration block, which just consists of sequential hopping, is included in the next exploration phase (lines 10-11 in Algorithm 3). No fixation phase is needed in that case.

### C.2.3   Communication phase

Notice that all active players are in a communication phase at the same time. However, this phase is decomposed into blocks of same length $T_d$ (to keep synchronization). A block can be of three different types, and the type of a block does not need to be the same for all players, as illustrated in Figure 2. Types are the following:

**Declaration block**  for player $j$: she communicates to other players that an arm is sub-optimal.

**Fixation block**  for player $j$: she tries to occupy any arm that she detected as optimal. If she succeeds, she exploits that arm until the end.

**Reception block**  for player $j$: she hops sequentially in order to detect other players' declarations.

Player $j$ starts the communication phase with declaration blocks, one per arm detected as sub-optimal[7]. She then proceeds to a fixation block, had she detected any arm as optimal during the last exploitation phase. She then proceeds to reception blocks until no new declaration is detected. As soon as no new declaration is detected, she starts a new exploration phase.

Notice that players keep receiving declarations from other players in any type of block.

**Declaration block:**   In a declaration block, player $j$ follows **Declare Protocol**, described in Pseudocode 7. The idea is to frequently sample the sub-optimal arm in order to send a "signal" to the other players. They will detect this signal by observing a significant loss in the empirical reward of this arm. However, a player sending a signal should also be able to detect signals on other arms sent by other players. That is the reason why in order to declare an arm as sub-optimal, a player randomly chooses between pulling this arm and sequentially hopping.

---

**Declare Protocol**

    **Input:** $k$ (arm to declare), $T_d$ (time of block), $\pi$ (first arm to pull in sequential hopping), $\mathbf{S}$ and $\mathbf{T}$ (exploration statistics), $[K_p]$ (active arms)

    **Output:** $D$ (signaled arms in this block)

1: Initialize $\mathbf{s}, \mathbf{t} \leftarrow \text{Zeros}(K)$
2: **for** $T_d$ time steps **do**
3:     Pull arm $i = \begin{cases} k \text{ with probability } \frac{1}{2} \\ \pi \text{ with probability } \frac{1}{2} \end{cases}$
4:     $s[i] \leftarrow s[i] + r_i(t); t[i] \leftarrow t[i] + 1$ and $\pi \leftarrow \pi + 1 \pmod{[K_p]}$
5: **end for**
6: $d \leftarrow$ set of active arms $i$ verifying $\left| \frac{S[i]}{T[i]} - \frac{s[i]}{t[i]} \right| \geq \frac{S[i]}{4T[i]}$
7: **return** $d \cup \{k\}$                                          *# arms signaled during the block*

---

Pseudocode 7:  Declare arm $k$ as sub-optimal.

Lemma 12 gives an appropriate choice for $T_d$ such that the declaration is detected by every player, without detecting any false positive declaration, no matter the block they are currently proceeding, with high probability.

Let $\hat{\mu}_i$ and $\hat{r}_i$ be respectively the empirical reward during the exploration phases and during the last communication block for arm $i$ and player $j$. Arm $i$ is detected as signaled, i.e., another player is declaring or exploiting this arm if:

$$|\hat{\mu}_i - \hat{r}_i| \geq \frac{\hat{\mu}_i}{4}. \tag{14}$$

Lemma 12 states that players will only detect arms declared as sub-optimal or exploited by a player with high probability. However, using the last reception block where there is no new signal, it is easy to distinguish exploited arms from declared ones. Indeed, for the exploited arms, only $0$ are observed during this last block; while for the declared ones, no player pulls it except for the sequential hopping. At least a positive reward will thus be observed with probability $1 - \frac{1}{T}$ on them during the block, thanks to its length $T_d$, which depends on $\mu_{\min}$.

**Fixation block:** In a fixation block, player $j$ proceeds to **Occupy Protocol**, described in Pseudocode 8. She sequentially hops on the active arms and starts exploiting an optimal arm as soon as it returns a positive reward (i.e., without collision at this step). In that case, she pulls this arm until the final horizon $T$. At the end of a block, if she did not manage to exploit any detected optimal arm, then all of them are occupied by other players with high probability thanks to the length of the block. Signals of other players are detected following the rule of Equation (14).

---

**Occupy Protocol**

    **Input:** $A$ (set to occupy), $T_d$ (time of block), $\pi$ (first arm to pull), $\mathbf{S}$ and $\mathbf{T}$ (exploration statistics), $[K_p]$ (active arms)
    **Output:** Fixed (exploited arm) , $D$ (signaled arms in this block)
1: Initialize $\mathbf{s}, \mathbf{t} \leftarrow \text{Zeros}(K)$; Fixed $\leftarrow -1$
2: **for** $T_d$ time steps **do**
3:     **if** Fixed $= -1$ **then**
4:         Pull $\pi$
5:         **if** $\pi \in A$ and $r_\pi(t) > 0$ **then** Fixed $\leftarrow \pi$ **end if**       # *no collision on optimal arm*
6:         $s[\pi] \leftarrow s[\pi] + r_\pi(t)$; $t[\pi] \leftarrow t[\pi] + 1$ and $\pi \leftarrow \pi + 1 \ (\text{mod } [K_p])$
7:     **else** Pull Fixed **end if**
8: **end for**
9: $d \leftarrow$ set of active arms $k$ verifying $\left| \frac{S[k]}{T[k]} - \frac{s[k]}{t[k]} \right| \geq \frac{S[k]}{4T[k]}$
10: **return** (Fixed, $d$)

---

Pseudocode 8: Try to start exploiting an arm among $A$.

**Reception block:** In a reception block, player $j$ sequentially hops and detects the signals of other players, following the rule of Equation (14). This corresponds to **Receive Protocol**, described in Pseudocode 9.

---

**Receive Protocol**

    **Input:** $T_d$ (time of block), $\pi$ (first arm to pull), $\mathbf{S}$ and $\mathbf{T}$ (exploration statistics), $[K_p]$ (active arms)
    **Output:** $D$ (signaled arms in this block), $\mathbf{s}$ and $\mathbf{t}$ (statistics of the block)
1: Initialize $\mathbf{s}, \mathbf{t} \leftarrow \text{Zeros}(K)$
2: **for** $T_d$ time steps **do**
3:     Pull $\pi$
4:     Update $s[\pi] \leftarrow s[\pi] + r_\pi(t)$; $t[\pi] \leftarrow t[\pi] + 1$ and $\pi \leftarrow \pi + 1 \ (\text{mod } [K_p])$
5: **end for**
6: $d \leftarrow$ set of active arms $k$ verifying $\left| \frac{S[k]}{T[k]} - \frac{s[k]}{t[k]} \right| \geq \frac{S[k]}{4T[k]}$
7: **return** $(d, \mathbf{s}, \mathbf{t})$

---

Pseudocode 9: Detect other players' declarations (and wait).

Notice that every active player will at least proceed to one reception block per communication phase (if she does not end up occupying an optimal arm). The last reception block is considered as the

block where no new signal is detected. This block is thus the same for all active players with high probability. Moreover, the arms giving $0$ reward during this last reception block are the optimal arms exploited by other players. This allows to distinguish exploited arms (which are optimal) from declared ones (which are sub-optimal). This distinction is described in Pseudocode 10. As a consequence, active players share a common set of active arms $[K_p]$ and number of active players $M_p$ at the end of each communication phase.

---

**Update**

   **Input:** Decl (declared arms), $s$ (statistics of last reception block), $[K_p]$ (set of active arms), $M_p$ (number of active players)

   **Output:** $[K_{p+1}]$ (updated set of active arms), $M_{p+1}$ (updated number of active players)

1: Opt $\leftarrow \{i \in \text{Decl} \mid s[i] = 0\}$
2: $[K_{p+1}] \leftarrow [K_p] \setminus \text{Decl}$ and $M_{p+1} \leftarrow M_p - \texttt{length}(\text{Opt})$
3: **return** $([K_{p+1}], M_{p+1})$

---

Pseudocode 10: Update the active sets at the end of a communication phase.

The complete description of SIC-MMAB2 is given in Algorithm 3 below.

---

**Algorithm 3** SIC-MMAB2 algorithm

    **Input:** $T$ (horizon), $\mu_{\min}$ (lower bound of means)

1: **Initialization Phase:**
2: Set $T_c \leftarrow \lceil \frac{\log(T)}{\mu_{\min}} \rceil$; $\pi \leftarrow$ MusicalChairs $([K], KT_c)$
3: $M_p \leftarrow$ Estimate_M_NoSensing $(\pi, T_c)$
4: Initialize $T_0 \leftarrow \lceil \frac{2400 \log(T)}{\mu_{\min}} \rceil$; Decl $\leftarrow \emptyset$; $T_d \leftarrow 0$; $[K_p] \leftarrow [K]$ and $\mathbf{S}, \mathbf{T}, \mathbf{s}, \mathbf{t} \leftarrow$ Zeros$(K)$

5: **for** $p = 1, \ldots, \infty$ **do**

6:     **Exploration Phase:**
7:     $T_{\text{expl}} \leftarrow K_p 2^p T_0$
8:     **if** length(Decl)$> 0$ **then**     *# there was a declaration in the previous phase so players need*
                                           *to reach an orthogonal setting among the new set of active arms*
9:         $\pi \leftarrow$ MusicalChairs $([K_p], K_p T_c)$
10:     **else** $\mathbf{S} \leftarrow \mathbf{S} + \mathbf{s}$; $\mathbf{T} \leftarrow \mathbf{T} + \mathbf{t}$ and $T_{\text{expl}} \leftarrow T_{\text{expl}} - T_d$     *# statistics of the last reception block*
11:     **end if**
12:     **for** $T_{\text{expl}}$ steps **do**                                        *# start exploration*
13:         Pull $\pi$; $S[\pi] \leftarrow S[\pi] + r_\pi(t)$; $T[\pi] \leftarrow T[\pi] + 1$ and $\pi \leftarrow \pi + 1 \pmod{[K_p]}$
14:     **end for**

15:     **Communication Phase:**                                   *# $B_s = \sqrt{\frac{2\log(T)}{s}}$ here*
16:     Initialize $T_d \leftarrow K_p T_0$ and Decl as empty set
17:     Rej $\leftarrow$ set of active arms $k$ verifying $\#\{i \in [K_p] \mid \frac{S[i]}{T[i]} - B_{T[i]} \geq \frac{S[k]}{T[k]} + B_{T[k]}\} \geq M_p$
18:     Acc $\leftarrow$ set of active arms $k$ verifying $\#\{i \in [K_p] \mid \frac{S[k]}{T[k]} - B_{T[k]} \geq \frac{S[i]}{T[i]} + B_{T[i]}\} \geq K_p - M_p$
19:     **while** Rej $\setminus$ Decl $\neq \emptyset$ **do**                                 *# declaration blocks*
20:         Let $k \in$ Rej $\setminus$ Decl
21:         $d \leftarrow$ Declare$(k, T_d, \pi, \mathbf{S}, \mathbf{T}, [K_p])$ and add $d$ to Decl
22:     **end while**
23:     **if** Acc $\setminus$ Decl $\neq \emptyset$ **then**                                   *# fixation block*
24:         (Fixed, $d$) $\leftarrow$ Occupy(Acc $\setminus$ Decl, $T_d, \pi, \mathbf{S}, \mathbf{T}, [K_p]$) and add $d$ to Decl
25:         **if** Fixed $\neq -1$ **then** go to line 35 (break) **end if**
26:     **end if**
27:     $d \leftarrow \{0\}$
28:     **while** $d \neq \emptyset$ **do**                                       *# reception blocks*
29:         $(d, \mathbf{s}, \mathbf{t}) \leftarrow$ Receive$(T_d, \pi, \mathbf{S}, \mathbf{T}, [K_p])$                *# $\mathbf{s}$ and $\mathbf{t}$ are the statistics*
30:         $d \leftarrow d \setminus$ Decl and add $d$ to Decl              *# so $d$ contains only the new signals.*
31:     **end while**

32:     **Update Statistics:**
33:     $([K_p], M_p) \leftarrow$ Update(Decl, $s$, $[K_p], M_p$)

34: **end for**

35: **Exploitation Phase:** Pull Fixed until $T$

---

## C.3   Regret analysis

This section is devoted to the proof of Theorem 3. It first proves several required lemmas.

A decomposition of the regret similar to SIC-MMAB is used for SIC-MMAB2:

$$R_T = R^{\text{init}} + R^{\text{explo}} + R^{\text{comm}}.$$

But in this section, a communication step is defined as a time step in a communication phase where there is at least a player using **Declare** or **Occupy protocol**, and $T_{\text{init}} := 3KT_c$. Notice that the last reception block of a communication phase then counts as communication steps only if there were declarations in previous blocks of the communication phase. Otherwise, its statistics are indeed used for the arms estimation, as described in Algorithm 3, lines 10-11, and it is then counted as exploration.

### C.3.1 Initialization regret

The initialization phase lasts $3KT_c$ steps, so $R^{\text{init}} \leq 3MKT_c$. Lemma 11 claims that the initialization is successful, meaning all players perfectly know $M$ after this phase, with a probability depending on $T_c$ and justifies the choice $T_c = \lceil \frac{\log(T)}{\mu_{\min}} \rceil$.

**Lemma 11.** *With probability* $1 - \mathcal{O}\left(MK \exp\left(-\mu_{\min} T_c\right)\right)$, *at the end of the initialization phase, every player has a correct estimation of $M$ and players are in an orthogonal setting.*

*Proof.* Similarly to the proof of Lemma 2, the probability to encounter a positive reward for a player during the Musical Chairs procedure at time step $t$ is lower bounded by $\frac{\mu_{\min}}{K}$. Hence using the same arguments, with probability $1 - \mathcal{O}\left(M \exp\left(-\mu_{\min} T_c\right)\right)$, all the players are pulling different arms after a time $KT_c$.

We now consider the **Estimate_M_NoSensing protocol**. Every time a player sends a bit to another player, it will be detected. Let us now bound the probability that a player detects a "collision" with another player while there is not. This is the case when she encounters $T_c$ successive zero rewards on an arm while she is the only player pulling it. This happens with probability smaller than $\exp\left(-\mu_{\min} T_c\right)$ for a single player in a single block. The union bound over the $M$ players and the $2K$ blocks yields the results. $\qquad\square$

### C.3.2 Communication regret

Lemma 12 provides the properties and regret of the algorithm during the communication phases.

**Lemma 12.** *Let the length of a block be such that $T_d = \lceil \frac{2400 K_p \log(T)}{\mu_{\min}} \rceil$, then conditionally on the successful outcome of all the previous Musical Chairs procedures:*

1. *with probability $1 - \mathcal{O}\left(\frac{M}{T}\right)$, a player $j$ declaring an arm $i$ as sub-optimal will be successfully detected by all active players;*

2. *with probability $1 - \mathcal{O}\left(\frac{MK}{T}\right)$, no player will detect a false signal during the declaration block (i.e., no arm is detected as declared if there was no declaration or if it is not occupied by an active player);*

3. *with probability $1 - \mathcal{O}\left(\frac{M}{T}\right)$, if player $j$ starts occupying arm $k$, it is detected as a declaration by all active players (following the rule of Equation (14)).*

*Thus, with probability $1 - \mathcal{O}\left(MK \exp(-\mu_{\min} T_c) + (K + \log(T))\frac{KM}{T}\right)$, all communication phases are successful, i.e., all signals are correctly detected and no false signal is detected. Then*

$$R^{\text{comm}} = \mathcal{O}\left(\frac{MK^2}{\mu_{\min}} \log(T)\right).$$

*Proof.* We first prove the three points conditionally on the success of the previous Musical Chairs procedures.

1) We prove this point in the more general case where the declaration of an arm $i$ follows the
   sampling: $\begin{cases} \text{Pull } i \text{ with probability } \lambda_d, \\ \text{Sample according to the sequential hopping on } [K_p] \text{ otherwise.} \end{cases}$

   First, denote by $T_{i'}^{j'}$ the number of pulls by player $j'$ on arm $i'$ during a block of length $T_d$. Using the Chernoff bound,

$$\mathbb{P}\left[T_{i'}^{j'} \leq \frac{(1-\lambda_d)T_d}{2K_p}\right] \leq \exp\left(-\frac{(1-\lambda_d)T_d}{8K_p}\right),$$
$$\leq \frac{1}{T} \qquad \text{as long as } \frac{(1-\lambda_d)T_d}{8K_p} \geq \log(T). \tag{15}$$

   This last condition holds with $T_d$ chosen as in Equation (16) and this inequality holds, no matter the type of block player $j'$ is proceeding.

With probability $1 - \mathcal{O}\left(\frac{KM}{T}\right)$, all the $T_i^j$ are thus greater than $\frac{(1-\lambda_d)T_d}{2K_p}$. This is also the case with probability 1 for the exploration pulls as the first exploration phase is of length $T_d$. Let $r_i$ and $\hat{r}_i$ respectively denote the expected and the empirical observed rewards of the arm $i$ during this declaration phase for player $j$. Assume that the arm $i$ is declared as sub-optimal by another player during the considered phase. It then holds that $r_i \leq \mu_i(1 - \lambda_d)$.

$$\text{With the specific choice of} \qquad T_d \geq \frac{300 K_p \log(T)}{(1-\lambda_d)\lambda_d^2 \mu_{\min}}, \qquad (16)$$

Chernoff bound provides the following inequalities, conditionally on $T_i^j \geq \frac{(1-\lambda_d)T_d}{2K_p}$,

$$\mathbb{P}\left[|\hat{r}_i - r_i| \geq \frac{\lambda_d \mu_i}{5}\right] \leq \frac{2}{T}$$

$$\text{and } \mathbb{P}\left[|\hat{\mu}_i - \mu_i| \geq \frac{\lambda_d \mu_i}{5}\right] \leq \frac{2}{T} \qquad \text{for the exploration phases.}$$

We then consider the high probability event $|\hat{r}_i - r_i| \leq \frac{\lambda_d}{5}\mu_i$ and $|\hat{\mu}_i - \mu_i| \leq \frac{\lambda_d}{5}\mu_i$.

As $\lambda_d \leq 1$, the second inequality yields $\frac{5}{6}\hat{\mu}_i \leq \mu_i \leq \frac{5}{4}\hat{\mu}_i$.

If $i$ is declared by a player, $\mu_i - r_i \geq \lambda_d \mu_i$ and

$$|\hat{\mu}_i - \hat{r}_i| \geq \frac{3\lambda_d}{5}\mu_i \geq \frac{\lambda_d}{2}\hat{\mu}_i. \qquad (17)$$

This means that with the detection rule described in Appendix C.2.3 for $\lambda_d = \frac{1}{2}$, for a single arm $i$ and player $j$, with probability $1 - \mathcal{O}\left(\frac{1}{T}\right)$, the player will correctly detect the declaration of arm $i$ as sub-optimal by (at least) another player.

2) As in the first point, with probability $1 - \mathcal{O}\left(\frac{1}{T}\right)$, it holds $|\hat{r}_i - r_i| \leq \frac{\lambda_d}{5}\mu_i$. The case of neither declaration nor exploitation by any other player actually corresponds to $r_i = \mu_i$. Thus we can rewrite Equation (17) of the first case into $|\hat{\mu}_i - \hat{r}_i| \leq \frac{2\lambda_d}{5}\mu_i \leq \frac{\lambda_d}{2}\hat{\mu}_i$, which holds with probability $1 - \mathcal{O}\left(\frac{1}{T}\right)$. Considering all arms and players yields the second point.

3) The same argument as in Lemma 11 gives that with probability $1 - \mathcal{O}\left(\frac{1}{T}\right)$, player $j$ will actually starts occupying $k$ after a time $t_{\text{fix}} \leq \frac{K_p \log(T)}{\mu_{\min}} \leq \frac{T_d}{2400}$.

Chernoff bound then provides a bound on the total reward $X_k^{j'}$ observed by $j'$ on $k$ for a time $t_{\text{fix}}$, i.e., for $T_k^{j'} \leq \frac{\log(T)}{\mu_{\min}}$ pulls on $k$.

$$\mathbb{P}\left[X_k^{j'} \geq 4\mu_k \frac{\log(T)}{\mu_{\min}}\right] \leq \exp\left(-\frac{3\mu_k \log(T)}{3\mu_{\min}}\right) \leq \frac{1}{T}. \qquad (18)$$

Thus, Equation (18) claims that with probability $1 - \mathcal{O}\left(\frac{1}{T}\right)$, $X_k^{j'} \leq \frac{4\mu_k \log(T)}{\mu_{\min}}$.

However, $k$ will be occupied after that point and no other positive reward will be observed by player $j'$. As a consequence, her empirical reward on $k$ will be for this block $\hat{r}_k^{j'} \leq \frac{2X_k^{j'} K_p}{(1-\lambda_d)T_d} \leq \frac{2\mu_k}{75}$. Using the same argument as in points 1) and 2), this guarantees $|\hat{\mu}_k^{j'} - \hat{r}_k^{j'}| \geq \frac{\hat{\mu}_k^{j'}}{4}$ and the result follows.

Conditionally on the success of the previous Musical Chairs procedures (i.e., players end these procedures in orthogonal settings), these three points imply that, with probability $1 - \mathcal{O}\left(\frac{MK}{T}\right)$, the communication block will be successful: all declarations are correctly detected, all detected optimal arms are exploited by a player and there is no false detection.

Let $N$ be the total number of exploration phases. By construction of the algorithm, $N \leq \lceil \log_2(T) \rceil$. Also there can not be two different blocks used to declare or occupy the same arm, conditionally on the

success of the previous communication blocks and Musical Chairs. Hence, conditionally on this event, there will be at most $N + K$ communication blocks, each succeeding with probability $1 - \mathcal{O}\left(\frac{KM}{T}\right)$ and there will be at most $K$ Musical Chairs procedures, each also succeeding with probability $1 - \mathcal{O}(M\exp(-\mu_{\min}T_c))$. Using a chain rule argument, all the communication protocols and Musical Chairs procedures are successful with probability $1 - \mathcal{O}\left(KM\exp(-\mu_{\min}T_c) + (K + N)\frac{KM}{T}\right)$ and the length of Comm is at most $\mathcal{O}\left(\frac{K^2\log(T)}{\mu_{\min}}\right)$, since only the communication phases with at least a Declaration or Fixation block are counted. This leads to the bound on $R^{\text{comm}}$ given by Lemma 12. $\quad\square$

### C.3.3 Exploration regret

Conditionally on the success of the initialization phase, all the communication phases and all the Musical Chairs procedures at the beginning of exploration phases, the exploration (except the Musical Chairs) will be collision-free. Using similar arguments as in Lemma 3, we provide an upper bound for the exploration regret of SIC-MMAB2.

**Lemma 13.** *With probability* $1 - \mathcal{O}\left(KM\exp\left(-\mu_{\min}T_c\right) + (K + \log(T))\frac{KM}{T}\right)$,

$$R^{\text{explo}} \leq \mathcal{O}\left(\sum_{k>M}\min\left\{\frac{M\log(T)}{\mu_{(M)} - \mu_{(k)}}, \sqrt{MT\log(T)}\right\} + \frac{MK^2}{\mu_{\min}}\log(T)\right).$$

*Proof.* First, as already claimed in the proof of the communication regret, the initialization, all the communication blocks and Musical chairs procedures succeed and there are at most $K$ Musical Chairs procedures during the exploration with probability $1 - \mathcal{O}\left(MK\exp(-\mu_{\min}T_c) + (K + \log(T))\frac{KM}{T}\right)$. The remaining of the proof is conditioned on this event. A single Musical Chairs procedure lasts a time $\frac{K_p\log(T)}{\mu_{\min}}$, hence the total regret incurred by the Musical Chairs is smaller than $\frac{MK^2\log(T)}{\mu_{\min}}$.

We now consider the regret of exploration without the Musical Chairs. We denote by $N$ the number of exploration phases that will be run and the same notation as in Appendix A.2.2 concerning $\Delta_k$. As the exploration phases are collision-free (conditionally on the success of initialization, communication and Musical Chairs), the Hoeffding inequality still holds: $\mathbb{P}\left[\exists p \leq n : |\hat{\mu}_k(p) - \mu_k| \geq \sqrt{\frac{2\log(T)}{T_k(p)}}\right] \leq \frac{2n}{T}$.

Since the players do not share their statistics, it can be shown with the same arguments as in Appendix A.2.2 that a sub-optimal arm $k$ will be found sub-optimal with probability at least $1 - \mathcal{O}\left(\frac{NM}{T}\right)$ after $t_k = \mathcal{O}\left(\frac{\log(T)}{\Delta_k^2}\right)$ exploration pulls **for a single player** without being found optimal by any player before. Since the exploration phases are collision-free, the cost for pulling the sub-optimal arm $k$ is $\mathcal{O}\left(\min\left\{\frac{M\log(T)}{\Delta_k}, \Delta_k T\right\}\right)$.

The same reasoning as in Appendix A.2.2 shows that the exploration regret due to non pulls of optimal arms is in $\mathcal{O}\left(\sum_{k>M}\min\left\{\frac{M\log(T)}{\mu_{(M)} - \mu_{(k)}}, \sqrt{MT\log(T)}\right\}\right)$ conditionally on correct estimations of the arms.

As $N \leq \lceil\log_2(T)\rceil$, all those arguments yield the bound on $R^{\text{explo}}$, with probability $1 - \mathcal{O}\left(\frac{KM\log(T)}{T} + KM\exp(-\mu_{\min}T_c) + (K + \log(T))\frac{KM}{T}\right)$. $\quad\square$

Theorem 3 can now be deduced from Lemmas 11, 12, 13 and Equation (1). The total regret is upper bounded by the sum of the regrets mentioned in Lemmas 11, 12, 13 and the regret when a "bad" event occurs. According to these lemmas, the probability that a bad event may happen is indeed in $\mathcal{O}\left((K + \log(T))\frac{KM}{T}\right)$. The average regret due to bad event is thus upper bounded by this probability multiplied by $MT$. This term is then dominated by the communication regret.

## D  On the inefficiency of SELFISH algorithm

A linear regret for the SELFISH algorithm in the No Sensing model has been recently conjectured [8]. This algorithm seems to have good results in practice, although rare runs with linear regret appear.

This is due to the fact that with probability $p > 0$ at some point $t$, both independent from $T$, some players might have the same number of pulls and the same observed average rewards for each arm. In that case, the players would pull the exact same arms and thus collide until they reach a tie breaking point where they could choose different arms thanks to a random tie breaking rule. However, it was observed that such tie breaking points would not appear in the experiments, explaining the linear regret for some runs. Here we claim that such tie breaking points might never happen in theory for the SELFISH algorithm when the rewards follow Bernoulli distributions, if we add the constraint that the numbers of positive rewards observed for the arms are all different at some stage. This event remains possible with a probability independent from $T$.

**Proposition 2.** *For $s, s' \in \mathbb{N}$ with $s \neq s'$:*

$$\forall n \geq 2, t, t' \in \mathbb{N}, \qquad \frac{s}{t} + \sqrt{\frac{2 \log(n)}{t}} \neq \frac{s'}{t'} + \sqrt{\frac{2 \log(n)}{t'}}.$$

*Proof.* First, if $t = t'$, these two quantities are obviously different as $s \neq s'$.

We now assume $\frac{s}{t} + \sqrt{\frac{2 \log(n)}{t}} = \frac{s'}{t'} + \sqrt{\frac{2 \log(n)}{t'}}$ with $t \neq t'$.

This means that $\sqrt{\frac{2 \log(n)}{t}} - \sqrt{\frac{2 \log(n)}{t'}}$ is a rational, i.e., for some rational $p$, $\log(n)(t + t' - 2\sqrt{tt'}) = 2p$.

It then holds
$$\log(n)\sqrt{tt'} = \log(n)\frac{t + t'}{2} - p,$$
$$tt' \log^2(n) = \log^2(n)(\frac{t + t'}{2})^2 - p(t + t') \log(n) + p^2,$$
$$\log^2(n)(\frac{t - t'}{2})^2 - p(t + t') \log(n) + p^2 = 0.$$

Since $(\frac{t-t'}{2})^2 \neq 0$ and all the coefficients are in $\mathbb{Q}$ here, this would mean that $\log(n)$ is an algebraic number. However, Lindemann–Weierstrass theorem implies that $\log(n)$ is transcendental for any integer $n \geq 2$. We thus have a contradiction. $\square$

The proof is only theoretical as computer are not precise enough to distinguish rationals from irrationals. The advanced arguments are not applicable in practice. Still, this seems to confirm the conjecture proposed by [8]: a tie breaking point is never reached, or at least not before a very long period of time.