[Reviews · NeurIPS 2019]

Reviewer 1



Originality: This paper invents the trick of implicit communication through collisions for synchronization case and it is used in subsequent papers. Also, it proposes the first algorithm without synchronization, which is an unrealistic assumption in real case. Quality: The proofs seem to be right and it clearly support the theorem given and the claim that the existing lower bound has potential problem. Clarity: The paper is well-organized. Significance: The paper gives a useful trick for synchronization case which uses the information deduced from collisions and explains why the previous lower bound has potential problem. It also invents the first algorithm with log regret in dynamic setting, which is more realistic.

Reviewer 2



Originality: The paper studies the multiplayer bandit problem and is largely based on the Musical Chairs paper (Rosenski et al.). The fact that the players can use the collisions to their advantage, and that the resulting algorithm enjoys a similar regret bound to the centralized setting, is nontrivial. Quality: The results are sound. Clarity: The paper is well-written overall. However, the authors do not explicitly address the fact that SIC-MMAB2 and its analysis are only found in the supplementary material. Significance: The theoretical results are important as they contradict two previously known lower bounds. --- After rebuttal --- I have read the authors feedback and am satisfied with the fixes planned for the camera-ready version. I wholeheartedly suggest the papers acceptance.

Reviewer 3



This paper deals with the problem of multi-player MAB, where the players either get collision information or do not have this information, and when the setting is static (i.e. all players start and finish at the same time) or dynamic (where players can join during the time horizon at different time points). The algorithms proposed in the paper are very interesting and innovative, and the ideas are original. The paper is written very well, clear and organised. The theoretical guarantees of the first algorithm have profound implication on the lower bounds of the problem of decentralized MMAB, showing that the existing lower bound are incorrect, and that synchronization is in some sense equivalent to centralized control (since it allows communication). The second algorithm which deals with the more complex setting, pf a dynamic and no-sensing MMAB problem, is able to achieve logarithmic regret and offer interesting ideas as well. I found the paper very significant and interesting. In terms of the technical details - I did not go over the proofs thus I cannot asses the correctness of the analysis. General Comments: - line 37 - In line 15 you are defining X_k(t) as the reward of arm k at time t. Then you say the X_k(t) is "additional information" and not just the regular reward information in the MAB. Since you are only defining r(t) later, this is confusing. - lines 155-156 - "Notice that players even use their own quantized statistics to accept/reject an arm instead of the exact ones. Otherwise, the sets of accepted or rejected arms could differ between the players" - this sentence seem to be self contradicting... I am not sure I understand this - if players use their own quantized statistics, how do they not end up with different accept/reject decisions? I suspect there is a mistake in the phrasing of this sentence. - line 196 - you mention experiments you performed for which the results are in the appendix, maybe you can state what they were (in a sentence, no need for more). - When you state that your regret is close to the lower bound on the centralized case - it will be helpful if you also state what is that lower bound. In addition, when stating that existing lower bounds for the decentralized case are incorrect - please also note what they are so that it is easy to see where is the gap. Technical Comments: - line 148 - "During," -> remove "," -- Added after author feedback -- I have read the author feedback and am satisfied with their response. I leave my score as is and advocate for acceptance.

[Author Response · NeurIPS 2019]

We first want to thank the reviewers for all their comments. You can be sure that they will all be taken into account in the rewriting process. We are glad to see that you all seem enthusiastic about this paper and its impact in the MMAB literature.

Overall, you seem to wish that some parts (especially Appendix C and the experiments) were a bit more detailed in the main text. You may have noticed that some choices were made due to space constraints, including postponing these two parts to the appendix. This choice was made because we believe that the whole part on SIC-MMAB2 is technically interesting, but that its impact is less significant than SIC-MMAB and DYN-MMAB (same goes for the experiments). However, the additional ninth page for the camera ready version would allow us to mention these parts in the main text by giving the main points of Appendix C and quickly describing the results of the experiments, besides taking into account all your other comments on the paper. We now specifically answer to each reviewer comments.

**Comments of reviewer 3:** As explained above, we should be able to add a description of the experiments and their results in the main text, and even maybe the figures.

**Comments of reviewer 4:** As for reviewer 3 with the experiments, we should be able to add a short description of the appendix C in the main text, as well as mentioning that Thm 3 and Eq 13 can be found there.

**Comments of reviewer 6:** All your comments will be taken into account in the paper.
- line 155-156 "Notice that players even use their own quantized statistics to accept/reject an arm instead of the exact ones. Otherwise, the sets of accepted or rejected arms could differ between the players". The current phrasing indeed seems confusing. What we mean is that when taking the decision of accepting/rejecting any arm, the player $j$ does not use its exact statistics $S_k^j(p)$ (which is more accurate) in the value of $\widetilde{\mu}_k^j(p)$ but instead uses the less accurate, quantized statistic $\widetilde{S}_k^j(p)$. Mathematically, we wanted to stress that $\widetilde{\mu}_k^j(p) = \frac{\sum_{m=1}^M \widetilde{S}_k^m(p)}{T_k(p)}$ and not $\frac{S_k^j(p) + \sum_{m \neq j} \widetilde{S}_k^m(p)}{T_k(p)}$.
By using $\widetilde{S}_k^j(p)$, the value of $\widetilde{\mu}_k^j(p)$ is the same for all $j$, whereas with the exact statistic $S_k^j(p)$ (only for $j$ but $\widetilde{S}_k^{j'}(p)$ for other $j'$), it would differ between players. It thus allows to have the same set of rejected/accepted arms among the players. We hope that this explanation makes it clearer for you. We will think of a better rephrasing for this sentence.

[Meta-Review · NeurIPS 2019]

The paper considers the multiplayer (stochastic) MAB problem in which M players compete over K > M arms, where the reward of an arm is sampled i.i.d. unless there is a collision (two or more players pull the same arm) in which case the reward is zero. The authors give new algorithms with regret bounds comparable to the best existing bounds for centralized algorithms. The reviewers unanimously agreed (and I concur) that the results are significant and the paper is well-written. A clear accept.